# Extraction, Isolation, and Purification of Value-Added Chemicals from Lignocellulosic Biomass

Tanmay Chaturvedi *, Laura Sini Sofia Hulkko , Malthe Fredsgaard and Mette Hedegaard Thomsen

Department of Energy, Aalborg University, 6700 Esbjerg, Denmark
* Correspondence: tac@energy.aau.dk

**Abstract:** This review covers the operating conditions for extracting top value-added chemicals, such as levulinic acid, lactic acid, succinic acid, vanillic acid, 3-hydroxypropionic acid, xylitol, 2,5-furandicarboxylic acid, 5-hydroxymethyl furfural, chitosan, 2,3-butanediol, and xylo-oligosaccharides, from common lignocellulosic biomass. Operating principles of novel extraction methods, beyond pretreatments, such as Soxhlet extraction, ultrasound-assisted extraction, and enzymatic extraction, are also presented and reviewed. Post extraction, high-value biochemicals need to be isolated, which is achieved through a combination of one or more isolation and purification steps. The operating principles, as well as a review of isolation methods, such as membrane filtration and liquid–liquid extraction and purification using preparative chromatography, are also discussed.

**Keywords:** value-added chemicals; lignocellulose biomass; pretreatment; extraction; isolation



## 1. Introduction

Lignocellulosic biomass is the most widely available feedstock for biofuel production. As a second-generation feedstock, lignocellulosic biomass does not compete with crops used for food products, such as corn, sugarcane, beetroot, and others. Traditionally, large-scale biomass processing facilities have focused on two main bio-based products, i.e., biofuels and bioenergy. Biofuels include ethanol, butanol, biodiesel, etc. [1,2]. Methane from biogas plants and syngas constitute the primary bioenergy products obtained from biomass processing [3,4]. Whereas these products have spurred the quest for cleaner fuels, they have fallen short of presenting a sustainable business model for production of high-value biomass-derived products. Moving away from single feedstock to a single-product approach leads toward the contemporary biorefinery approach, whereby multi feedstock processing leads to multiple bio-based products, including but not limited to biofuel, bioenergy, biochemicals, proteins, and other high-value bioproducts.

In 2004, the U.S. Department of Energy (USDOE) identified the top 12 platform chemicals (based on their market potential) that can be derived from biomass [5]. Four carbon (C4) 1,4-dicarboxylic acids (succinic, fumaric, and malic acid), 2,5-furandicarboxylic acid, 3-hydroxypropionic acid, aspartic acid, glucaric acid, glutamic acid, itaconic acid, 3-hydroxybutyrolactone, glycerol, sorbitol, and xylitol were identified as the top 12 candidates that can subsequently be converted into numerous high-value biomass-derived products. 'Top Value Added Chemicals from Biomass' was the first volume of a two-volume report, with the second volume published in 2007, in which the list was updated to include lignin-based derivatives, such as vanillin, vanillic acid, syringaldehyde, aromatic diacids, and quinones, amongst others [6]. Several promising technologies to obtain these building block/intermediate chemicals are identified in this report based on technological maturity and reported product yields.

## 2. Methodology

In this paper, we build on the foundation laid by these two reports and aim to consolidate the literature with respect to the effectiveness of extracting some of these value-added

chemicals from the most common lignocellulosic biomasses. The most common lignocellulosic biomasses include corn stover, sugarcane bagasse, pine, wheat straw, rice straw, softwood, aspen wood, etc. This review identifies which value-added chemicals can be derived from the most common lignocellulosic biomass and presents their respective yields. Furthermore, we evaluate the technologies and the operating conditions used to extract value-added chemicals from the most common lignocellulosic biomasses. Subsequently, methods for the isolation of chemicals are also discussed.

Three factors govern the processing of lignocellulosic biomass: availability of the type of biomass, accessibility and maturity of processing technology, and the intended bioproduct and its market demand. We begin this review begins by identifying the critical value-added chemicals based on their availability and potential to be extracted from lignocellulosic biomass. Next, the literature-reported concentrations, yield, and productivity of extracting value-added chemicals from some of the most common lignocellulosic biomasses are presented. Lastly, the most promising technologies for the isolation of bioproducts are reported.

Lignocellulosic biomass can be broken down into three major building blocks: cellulose, hemicellulose, and lignin. Value-added chemicals can be categorized according to their functional groups of origin, and based on processing routes as shown in Figure 1.

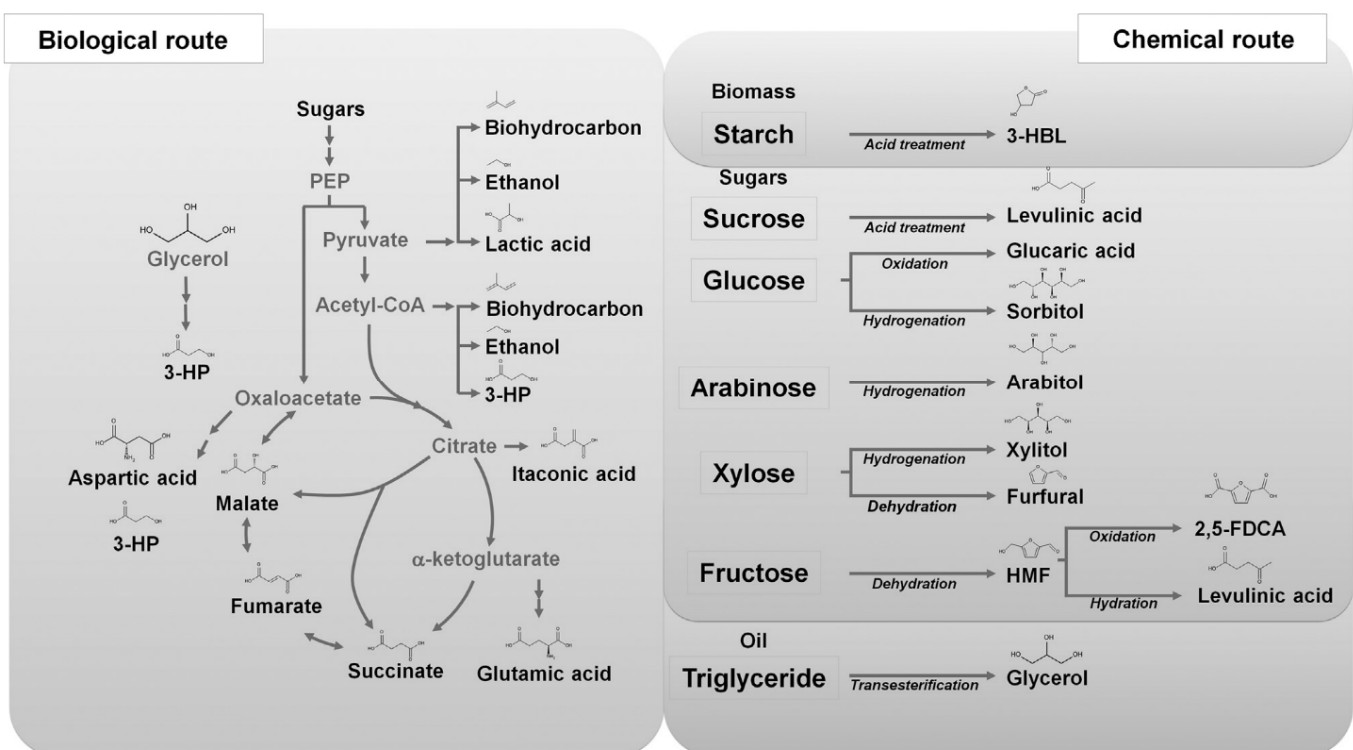

**Figure 1.** Biological and chemical routes for the production of top value-added chemicals derived from lignocellulosic biomass. Reprinted/adapted with permission from Ref. [7].

In this paper, we will discuss the following derivatives of cellulose:

1. Levulinic acid;
2. Lactic acid;
3. 3-hydroxypropionic acid (3-HP);
4. Succinic acid;
5. Vanillic acid and vanillin;
6. Itaconic acid;
7. Adipic acid;
8. 2,5-furandicarboxylic acid (FDCA); and
9. 5-hydroxymethylfurfural (HMF).

The value-added chemicals extracted from hemicellulose and lignin discussed in this paper are:

1. Xylitol;
2. Furfural;
3. Chitosan;
4. 2,3-butanediol (2,3-BD); and
5. Xylo-oligosaccharides (XOs)

## 3. Top Value-Added Chemicals

### 3.1. Levulinic Acid

Levulinic acid can be extracted by the dehydration of sugars, hydration of hydroxymethylfurfural (HMF) or hydrolysis of furfuryl alcohol, both of which are derived from xylose, which is a hemicellulose sugar [5]. A simple structure of levulinic acid is seen in Figure 2. Levulinic acid has applications in additives, pharmaceutical, and plastic industries [7]. Biofine Technology (Boston, MA, USA), GFBiochemicals (Paris, France), and Avantium (Amsterdam, The Netherlands) are companies that are invested in the commercial production of levulinic acid. Table 1 summarizes the levulinic acid yield extracted from lignocellulosic biomass under optimized pretreatment conditions (acid concentration, time, and operating temperature). Levulinic acid can serve as a precursor for succinic acid, diphenolic acid, valeric acid, γ-valerolactone, acetyl acrylic acid, 1,4-butanediol, and other value-added chemicals [8,9]. The ease of deriving levulinic acid from a variety of lignocellulosic crops and its important position in the supply chain as an intermediate for the production of resins, herbicides, plasticizers, solvents, fuels, food, flavoring, and fragrance components makes it one of the top value-added chemicals that can be derived in a lignocellulosic biorefinery.

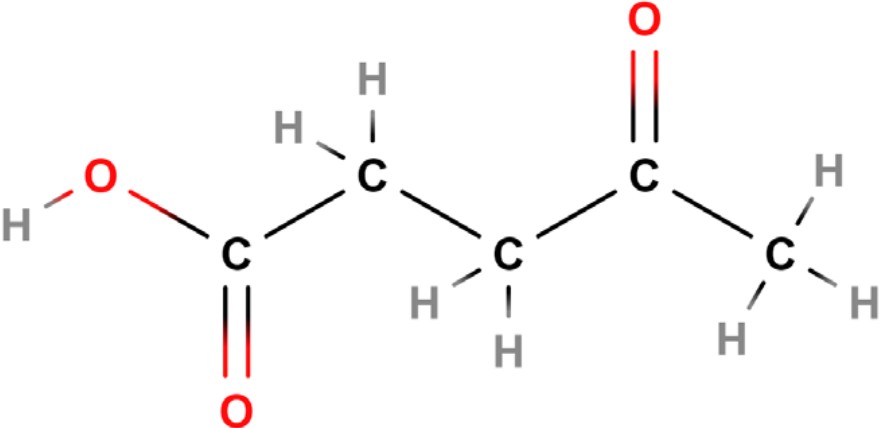

**Figure 2.** Structure of levulinic acid ($C_5H_8O_3$).

Table 1. Review of extraction methods for levulinic acid from lignocellulosic biomass.

| Biomass | Cellulose Content % | Acid Concentration | Operating Temperature (°C) | Time (h) | Theoretical Yield (mol%) | Reference |
|---|---|---|---|---|---|---|
| Kraft Paper Pulp Residue | 80 | 1–5% $H_2SO_4$ | 1st Stage 210–230 2nd Stage 195–215 | N/A | 70–80 | [10] |
| | 40 | 3.5% $H_2SO_4$ | 210 | N/A | 68.8 | [11] |
| Wheat Straw | 40.4 | 4.5% $H_2SO_4$ | 220 | N/A | 79.6 | [12] |
| | | 3.5% $H_2SO_4$ | 210 | 0.63 | 19.8 | [13] |
| Bagasse | | 4.5% HCl | 220 | N/A | 82.7 | [12] |
| | 42 | 1.5% $H_2SO_4$ | 25–195 | 2 | 17.5 | [14] |
| | 32 | 20% HCl | 100 | 24 | 15 | [15] |
| | 29 | 6.5% HCl | 162 | 1 | 24 | [16] |
| | 27 | Amberlite IR-120 | Room Temperature | 124 | 5.8 | [17] |
| Glucose | 5–20 | 0.1–4% $H_2SO_4$ | 160–240 | N/A | 35.4 | [18] |
| | 10 | 6% HCl | 160 | 0.25 | 41.4 | [19] |
| | 12 | 3% Clay Catalyst (Fe-pillared montmorillonite) | 150 | 24 | 12 | [20] |
| | 12 | 3% HY Zeolite | 150 | 24 | 6 | [21] |
| Rice Hull | N/A | 1% HCl | 160 | 3 | 10.3 | |
| Rice Straw | N/A | 1% HCl | 160 | 3 | 5.5 | [22] |
| Corn Stalks | N/A | 1% HCl | 160 | 3 | 7.5 | |
| Wood Sawdust | N/A | 1.5% HCl | 190 | 0.5 | 9 | [23] |
| Oakwood | N/A | 3% $H_2SO_4$ | 180 | 3 | 17.5 | [24] |
| Aspen, Pine, and Spruce | N/A | 5% $H_2SO_4$ | 200–240 | 2–4 | 13–18 | [13] |
| Cellulose | N/A | 1–5% $H_2SO_4$ 1–5% HCl 1–5% HBr | 150–250 | 2–7 | <25 <28 <27 | [25] |
| Aspen Wood | N/A | 1–5% $H_2SO_4$ 1–5% HCl 1–5% HBr | 150–250 | 2–7 | <15.5 <12.4 <13 | |

### 3.2. Lactic Acid

Lactic acid (structure in Figure 3) has become increasingly popular as a biomass-derived chemical due to its utility for production of polylactic acid (PLA). Currently, Corbin (Amsterdam, The Netherlands), Futerro (Escanaffles, Belgium), NatureWorks (Minnetonka, MN, USA), and Myriant (Quincy, MA, USA) are commercial-scale users of lactic acid [7]. Some of the most common lactic acid bacteria used for lactic acid production from lignocellulosic-derived sugars include *L. planterum*, *L. pentosus*, *L. delbrueckii*, *L. casei*, *L. brevis*, *E. mundtii*, *E. faecalis*, *L. coryniformis*, *L. rhamnosus*, *L. salivarius*, *L. amylovorans*, and *L. amylophilus*, amongst others [26]. Lactic acid also serves as a critical platform chemical for production of lactide (the intermediate for polylactic acid production), propanoic acid, 1,2-propanediol, polyurethanes, pyruvic acid, acrylic acid, 2,3-pentanedione, and others. Lactic acid concentration, yield, and productivity from some of the most common lignocellulosic biomasses are shown in Table 2.

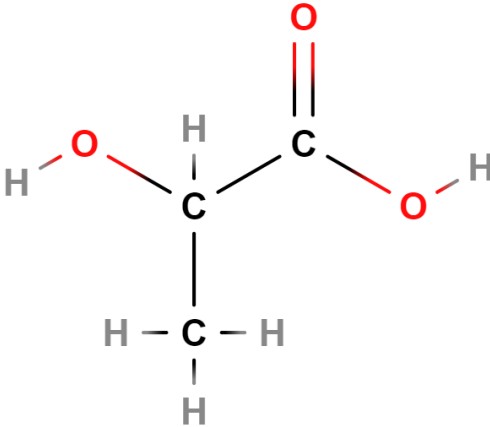

**Figure 3.** Structure of Lactic Acid ($C_3H_6O_3$).

**Table 2.** Lactic acid's concentration, yield, and productivity from some of the most common lignocellulosic biomasses.

| Biomass | Strain | Concentration (g/L) | Yield [a] (g/g) | Productivity [b] (g/L/h) | Reference |
|---|---|---|---|---|---|
| Wood Hydrolysate | *E. mundtii* QU 25 | 93 | 0.93 | 1.7 | [27] |
| Corn Cob/Stover | *Lb. brevis* | 39.1 | 0.7 | 0.81 | [28] |
| | *L. delbrueckii* ZU-S2 | 48.7/44.2 | 0.95/0.92 | 1.01/5.7 | [29] |
| | *L. pentosus* | 26 | 0.53 | 0.34 | [30] |
| | *L. pentosus* ATCC 8041 | 74.8 | 0.65 | N/A | [31] |
| | *L. rhamnosus* and *L. brevis* | 20.95 | 0.7 | 0.58 | [32] |
| Wheat Straw | *L. brevis* and *L. pentosus* | 7.1 | 0.95 | N/A | [33] |
| Softwood | *L. casei* subsp. *rhamnosus* | 21.1–23.75 | 0.74–0.83 | 0.15–0.23 | [34] |
| Sugarcane Bagasse | *L. delbrueckii* subsp. *delbrueckii* Mutant Uc-3 | 67 | 0.83 | 0.93 | [35] |
| | *L. lactis* IO-1 | 10.9 | 0.36 | 0.17 | [36] |
| Rice and Wheat Barn | *L. rhamnosus* ATCC 9595 (CET288) | 129 | 0.95 | 2.9 | [37] |
| Brewer's Spent Grain | *L. delbrueckii* UFV H2B20 | 35.5 | 0.99 | 0.59 | [38] |

[a] Ratio of the yield of lactic acid produced (g) to substrate consumed (g). [b] Lactic acid productivity.

### 3.3. 3-Hydroxypropionic Acid

3-hydroxypropionic acid (3-HP) is an important C3 platform chemical, primarily due to its contribution as precursor for the production of 1,3-propanediol. 3-HP is also a platform chemical used for the production of malonic acid, acrylic acid, acrylonitrile, polyamides, and 3-hydroxypropionate esters and its structure can be seen in Figure 4. BASF-Cargill-Novozymes (Ludwigshafen, Germany/Wayzata, MN, USA/Bagsværd, Denmark), and Dow (Midland, MI, USA) are commercial-scale producers of 3-HP [7].

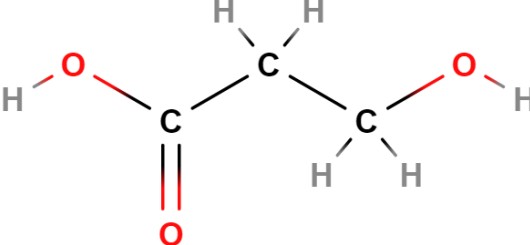

**Figure 4.** Structure of 3-hydroxypropionic acid ($C_3H_6O_3$).

3-HP can be produced via microbial fermentation pathways from two major substrates: glucose and glycerol [39]. Matsakas et al. [40] presented a comprehensive review on 3-HP production and its production pathway mechanisms. Table 3 provides an overview of how various microorganism compare in terms of their concentrations and productivity of 3-HP, with glycerol and glucose as the two main substrate sources.

**Table 3.** 3-hydroxypropionic acid concentration and productivity based on various microbes and two main substrate sources, i.e., glucose and glycerol.

| Substrate | Host Microorganism | Concentration (g/L) | Productivity (g/L/h) | Reference |
|---|---|---|---|---|
| Glucose | S. cerevisiae | 9.8 | 0.1 | [41] |
| | | 13.7 | 0.17 | [42] |
| | | 7.4 | 0.06 | [43] |
| | E. coli | 10.1 | 0.28 | [44] |
| | | 40.6 | 0.56 | [45] |
| | | 31.1 | 0.63 | [7] |
| | | 29.7 | 0.54 | [46] |
| | S. pombe | 7.6 | 0.25 | [47] |
| | C. glutamicum | 62.6 | 0.87 | [48] |
| Glycerol | K. pneumoniae | 18 | 0.77 | [49] |
| | | 48.9 | 1.75 | [50] |
| | | 43 | 0.9 | [51] |
| | | 83.8 | 1.16 | [52] |
| | | 0.9 | 0.04 | [53] |
| | | 24.4 | 1.02 | [50] |
| | | 16 | 0.3 | [54] |
| | | 11.3 | 0.94 | [55] |
| | | 22.7 | 0.38 | [56] |
| | | 28.1 | 0.58 | [57] |
| | | 22 | 0.46 | [58] |
| | | 60.5 | 1.12 | [59] |
| | E. coli | 42.1 | 1.32 | [60] |
| | | 71.9 | 1.8 | [61] |
| | | 40.5 | 1.35 | [62] |
| | | 56.4 | 1.18 | [63] |
| | | 41.5 | 0.86 | [64] |
| | | 31 | 0.43 | [65] |
| | | 38.7 | 0.54 | [66] |
| | | 6.06 | 0.13 | [67] |
| | | 5.05 | 0.105 | [68] |
| | L. reuteri | 10.6 | 1.08 | [69] |
| | | 3.3 | 0.09 | [70] |
| | L. collinoides | 0.55 | 0.07 | [71] |

### 3.4. Succinic Acid

Succinic acid is a four-carbon dicarboxylic acid (structure in Figure 5) that has been produced via chemical routes in the past but is gaining popularity for production via the biological route. Myriant Technologies now PTT Global Chemical (Bangkok, Thailand) and Reverdia (Utrecht, the Netherlands) have commercial-scale facilities for the production of bio-based succinic acid [7].

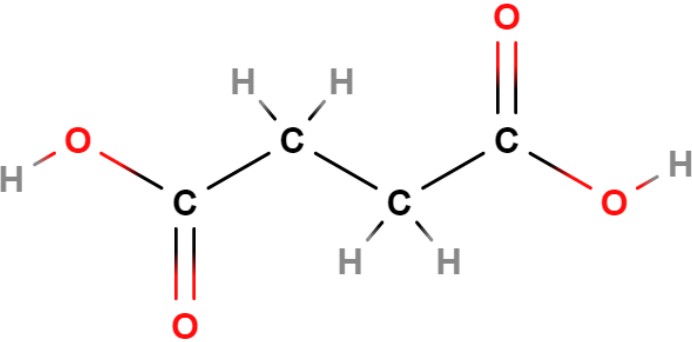

**Figure 5.** Structure of succinic acid ($C_4H_6O_4$).

Succinic acid is a key intermediate chemical used for the production of several derivatives of industrial importance, such as 1,4-butanediol, tetrahydrofuran, N-methylpyrrolidone, and γ-butyrolactone. These derivatives are, in turn, utilized in the production of polyurethanes, polyesters, and polyvinylpyrrolidone (PVP). Succinic acid can be extracted from various lignocellulosic biomasses. Table 4 provides an overview of succinic acid concentrations, alongside the concentrations of other value-added chemicals that can be derived simultaneously under varying pretreatment conditions, from some of the most common lignocellulosic biomasses.

### 3.5. Vanillic Acid and Vanillin

Vanillin is a widely used food flavoring agent that is most typically extracted from *Vanilla* spp.; however, it is currently produced inexpensively via petrochemical routes (structure of vanillic acid and vanillin in Figure 6). Besides the food industry, vanillin finds applications in the pharmaceutical and fragrance industries. The increasing demand for this molecule has propelled the search for biomass-derived pathways for vanillin production.

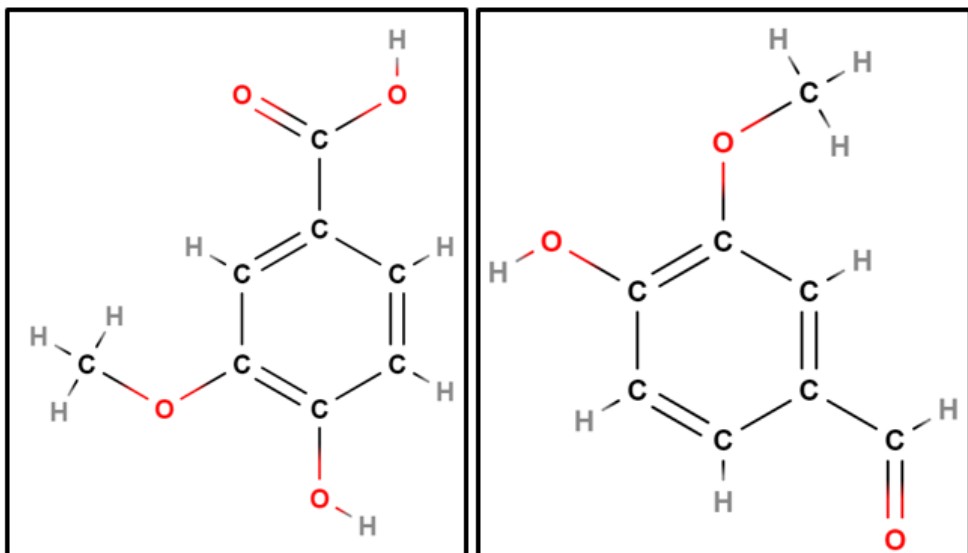

**Figure 6.** Structure of vanillic acid ($C_8H_8O_4$) (**left**) and vanillin ($C_8H_8O_3$) (**right**).

Vanillin can be extracted from lignin, ferulic acid, glucose, vanillic acid, aromatic amino acids, isoeugenol, waste residue, and other substrates [72]. Currently, Borregaard (Sarpsborg, Norway), a Norwegian company, claims to be the only producer of biovanillin, which it produces from wood. Table 4 summarizes some lignocellulosic biomasses that can be used to derive vanillin and vanillic acid.

### 3.6. Itaconic Acid

Itaconic acid is a C5 dicarboxylic acid (see Figure 7) that does not have as large a market share as the likes of succinic acid, levulinic acid, and lactic acid. However, it remains an intermediate building block of interest due to its significance in producing other value-added chemicals with larger market shares. Itaconic acid is a precursor for polymethyl methacrylate (PMMA), 3-methyltetrahydrofuran, polyitaconic acid, and styrene-butadiene rubber latex. Additionally, itaconic acid can be converted to methyl pyrrolidones, 2-methylbutanediol, 3-methyltetrahydrofuran, 4-methyl-γ-butyrolactone, and 4-methyl-γ-butyrolactone. Itaconic acid is recommended as a replacement for maleic acid/anhydride and sodium tripolyphosphate, which are, in turn, used for the production of polyester resin and detergent [5].

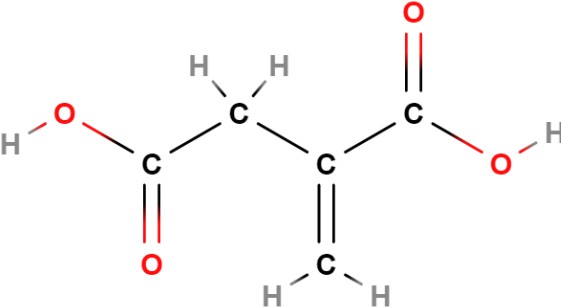

**Figure 7.** Structure of itaconic acid ($C_5H_6O_4$).

Qingdao Kehai Biochemistry (Jiaonan City, China), Zhejiang Guoguang Biochemistry (Quzhou City, China), Jinan Huaming Biochemistry (Mingshui Zhangqiu City, China), and Itaconix (Stratham, NH, USA) are producers of itaconic acid on a commercial scale [7]. Table 4 provides a summary of lignocellulosic biomasses and their respective pretreatment conditions for extraction of itaconic acid.

### 3.7. Adipic Acid

Adipic acid is a C6 dicarboxylic acid (see Figure 8) with applications in the production of nylon -6,6 fibers, resins, plasticizers, polyester polyols, food ingredients, and lubricants [5].

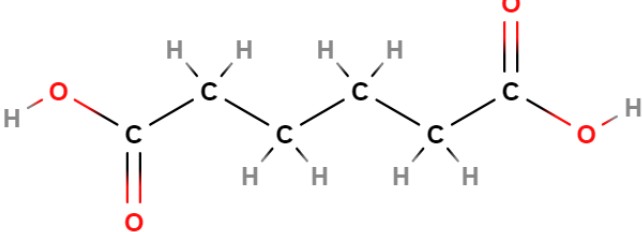

**Figure 8.** Structure of adipic acid ($C_6H_{10}O_4$).

Adipic acid is also a precursor for the production of muconic acid and glucaric acid. Biomass-derived adipic acid can be obtained from glucose as a starting substrate and *E. coli*, *S. cerevisiae*, and *P. putida* as the chassis [73]. Lignin, lipids, xylose, and amino acids can also serve as substrates to obtain bio-derived adipic acid. Genomatica (San Diego, CA, USA) and DSM (Heerlen, The Netherlands) are developing strategies to produce commercial-scale quantities of biomass-derived adipic acid [74].

### 3.8. Furfural

Furfural is a C5 molecule (as shown in Figure 9) most often derived from the hemicellulose fraction in lignocellulosic biomass. Furfural is a top value-added chemical with several commercial-scale facilities operating across China, South Africa, and the Dominican Republic [75]. The largest fractions of furfural are used in the production of furfuryl alcohol, which, in turn, is used in resin production.

**Figure 9.** Structure of furfural ($C_5H_4O_2$).

Furfural is the precursor for the production of furoic acid, tetrahydrofuran, fumaric acid, 2-methyltetrahydrofuran, tetrahydrofurfuryl alcohol, and furfurylamine [7]. The highest reported furfural yields from lignocellulosic biomass vary from 34 to 87 weight% (wt%) [76–78]. Table 4 highlights the concentrations of furfural obtained after pretreatment from some of the most common lignocellulosic biomasses.

### 3.9. 5-Hydroxymethylfurfural (HMF)

5-hydroxymethylfurfural (HMF) is a C6 derivative obtained through dehydration of glucose and fructose, a structure of which can be seen in Figure 10 [79]. For lignocellulosic biomasses, cellulose is the main contributor to HMF production. HMF production, in principle, is possible from all biomasses containing hexoses and its oligomers, providing a wide range of possible feedstock for HMF production [75].

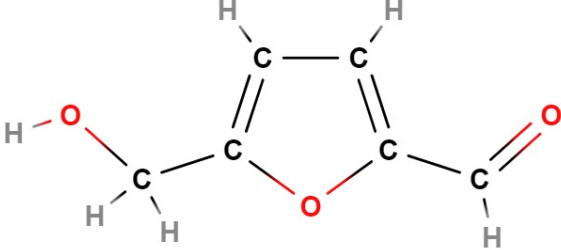

**Figure 10.** Structure of 5-HMF ($C_6H_6O_3$).

HMF is a precursor for the production of adipic acid, levulinic acid, 2,5-dimethylfuran, caprolactone, polyamide 6, 2,5-furandicarboxylic acid, and others. Despite the presence of hydroxyl and aldehyde functional groups in this molecule, industrial-scale production of HMF remains unfeasible, predominantly due to the high costs of fructose and low reactivity of cellulose. AVA Biochem (Muttenz, Switzerland), is one of the few companies close to commercial-scale production of HMF [7]. Table 4 shows HMF concentrations obtained for different lignocellulosic biomasses after pretreatment.

Table 4 provides an overview of succinic acid, glutaric acid, vanillin, vanillic acid, itaconic acid, adipic acid, furfural, and 5-HMF concentrations, alongside the concentrations of other value-added chemicals that can be derived simultaneously under varying pretreatment conditions from some of the most common lignocellulosic biomasses. Table 4 does not present the highest possible concentrations of each biochemical but, rather, presents the biorefinery view, wherein multiple value-added products can be obtained from lignocellulosic biomass.

**Table 4.** Pretreatment conditions and concentrations of selected furans, organic acids, and phenolics.

| Biomass | Feedstock Content | Pretreatment/ Catalyst | Operating Temp. (°C) | Time (min) | Furans | | | Organic Acids | | | | | Phenolics | | | Unit | Ref. |
|---|---|---|---|---|---|---|---|---|---|---|---|---|---|---|---|---|---|
| | | | | | Furoic Acid | Furfural | 5 HMF | Lactic Acid | Succinic Acid | Glutaric Acid | Itaconic Acid | Adipic Acid | Vanillin | Vanillic Acid | Ferulic Acid | | |
| Corn Stover | 34.4% glucan, 22.8% xylan, and 11% lignin | 0.7% $H_2SO_4$ (w/w) | 180 | 8 | 2.4 | 220 | 44 | 20 | 2.9 | 0.57 | 7.2 | 0.11 | 4 | 3.3 | 6.6 | g/L | [80] |
| | | 0.07% $H_2SO_4$ (w/w) | | | 1.1 | 26 | 11 | 17.8 | 1.7 | 0.24 | 2 | 0.14 | 2.8 | 1.5 | 2.6 | g/L | |
| | | Liquid hot water | | | 0.88 | 8 | 2.3 | 5.5 | 2.2 | 0.23 | 1.2 | 0.15 | 2.6 | 2.6 | 2.2 | g/L | |
| | | Deionized water saturated with oxygen at 174 psi | | | 1.2 | 6.5 | 2.8 | 24 | 5.2 | 0.65 | 2.1 | 0.2 | 6.7 | 4.3 | 1 | g/L | |
| | | Aqueous ammonia 0.1% (w/w) | | | 1.1 | 0.4 | 0.89 | 38 | 6.5 | 1.2 | 3.2 | 0.18 | 2.6 | 3.2 | 4.2 | g/L | |
| | 34.4% glucan, 22.4% xylan, 4.2% arabinan, 0.6% mannan, 1.4% galactan, 11% lignin, 2.3% protein, 6.1% ash, and 3.8% uronic acids | Ammonia fiber expansion (AFEX) | 130 | 15 | 0.006 | 0.003 | 0.642 | 0.318 | 0.596 | 0.008 | 0.022 | 0.003 | 0.195 | 0.046 | 0.103 | g/g DM | [81] |
| | | 30% $H_2SO_4$ (w/w) | 190 | | 0.155 | 7.94 | 15.7 | 1.5 | 0.26 | 0.012 | 0.58 | 0.005 | 0.281 | 0.124 | 1.314 | g/g DM | |
| | N/A | 1% $H_2SO_4$ (w/w) | 160 | 8 | N/A | 18.7 | 0.701 | 41 | N/A | N/A | N/A | N/A | 0.06 | 0.034 | N/A | mM | [82] |
| Poplar | 43.8% glucan, 14.85% xylan, 3.94% mannan, and 29.12% lignin | 0.7% $H_2SO_4$ (w/w) | 180 | 8 | 3.1 | 220 | 64 | 29 | 2.5 | 0.61 | 0.11 | 0.057 | 5.5 | 5.9 | 0.19 | g/L | [80] |
| | | 0.07% $H_2SO_4$ (w/w) | | | 1.7 | 31 | 4 | 19 | 0.93 | 0.26 | 0.13 | 0.1 | 5.6 | 5.7 | 0.46 | g/L | |
| | | Liquid hot water | | | 0.94 | 2.6 | 0.45 | 1.8 | 2.3 | 0.23 | 0.093 | 0.048 | 3.1 | 4.1 | 0.23 | g/L | |
| | | Deionized water saturated with oxygen at 12 bar | | | 0.76 | 2.1 | 0.39 | 22 | 2.4 | 0.25 | 0.17 | 0.14 | 9.1 | 5.3 | 0.07 | g/L | |
| | | Aqueous ammonia 0.1% (w/w) | | | 0.49 | 0.5 | 0.079 | 26 | 1.7 | 0.35 | 0.088 | 0.13 | 2.8 | 2.5 | 0.13 | g/L | |
| | 48.9% glucan, 15.7% xylan, 27.7% lignin, and 1.2% ash | Steam explosion | 214 | 6 | N/A | 5.9 | 2.6 | N/A | N/A | N/A | N/A | N/A | 0.035 | N/A | N/A | mg/g DM | [83] |

**Table 4.** *Cont.*

| Biomass | Feedstock Content | Pretreatment/ Catalyst | Operating Temp. (°C) | Time (min) | Furans | | | Organic Acids | | | | | Phenolics | | | Unit | Ref. |
|---|---|---|---|---|---|---|---|---|---|---|---|---|---|---|---|---|---|
| | | | | | Furoic Acid | Furfural | 5 HMF | Lactic Acid | Succinic Acid | Glutaric Acid | Itaconic Acid | Adipic Acid | Vanillin | Vanillic Acid | Ferulic Acid | | |
| Pine | 40% glucan, 8.9% xylan, 16% mannan, and 27.7% lignin | 0.7% $H_2SO_4$ (*w/w*) | 180 | 8 | 1.1 | 190 | 170 | 3.7 | 0.73 | 0.37 | 0.07 | 0.076 | 4.6 | 5.2 | 0.12 | g/L | [80] |
| | | 0.07% $H_2SO_4$ (*w/w*) | | | 0.8 | 13 | 9.5 | 4.5 | 0.34 | 0.18 | 0.032 | 0.09 | 5.8 | 3.6 | 0.22 | g/L | |
| | | Liquid hot water | | | 0.83 | 2.5 | 1.3 | 8.7 | 0.75 | 0.16 | 0.09 | 0.054 | 2.4 | 2.3 | 0.31 | g/L | |
| | | Deionized water saturated with oxygen at 12 bar | | | 0.91 | 1.9 | 0.64 | 18 | 1.8 | 0.31 | 0.24 | 0.18 | 7.1 | 4.8 | 0.14 | g/L | |
| | | Aqueous ammonia 0.1% (*w/w*) | | | 0.55 | 0.65 | 0.16 | 36 | 2.39 | 0.66 | 0.099 | 0.13 | 3.2 | 4.8 | 0.16 | g/L | |
| Spruce | 41.6% glucan, 11.5% mannan, 4.7% xylan, 2% galactan, 1.1% arabinan, 25.7% lignin, and 5.4% extractives | 0.5% $H_2SO_4$ (*w/w*) | 222 | 7 | N/A | 1 | 5.9 | N/A | N/A | N/A | N/A | N/A | 0.12 | 0.034 | N/A | g/L | [84] |
| Wheat Straw | 36.3% cellulose, 30.9% hemicellulose, and 7.1% lignin | 6.5g/L $Na_2CO_3$ | 185 | 10 | N/A | N/A | N/A | 0.461 | 0.899 | N/A | N/A | N/A | 0.008 | 0.004 | 0.009 | g/100 g DM | [85] |
| | | 2g/L $Na_2CO_3$ | 195 | 15 | 0.017 | 0.146 | 0.016 | N/A | 0.447 | N/A | N/A | N/A | 0.096 | 0.084 | 0.015 | | |
| Barley Straw | 33% glucan, 20% xylan, 3.8% arabinan, 1% galactan, 16.1% lignin, 7.6% ash, and 13.8% extractives | N/A | 210 | 5 | N/A | 0.28 | 0.08 | N/A | N/A | N/A | N/A | N/A | 25 | 4.4 | 10 | mg/100 g DM | [86] |

DM: dry matter. Adapted from [87].

### 3.10. 2,5-Furandicarboxylic Acid

2,5-Furandicarboxylic acid (FDCA) is considered a promising alternative for petroleum-derived terephthalic acid for the production of bioplastics, such as polyamides, polyesters, and polyethylene furandicarboxylate. FDCA can be synthesized from 5-HMF or 2-furoic acid derived from lignocellulose-based C6 and C5 sugars, respectively. The structure of FDCA can be seen in Figure 11. Besides the required pretreatment for lignocellulose saccharification, the low efficiency of the dehydration process from hexoses to 5-HMF has been an obstacle for commercial production [88,89]. For example, reported yields from the conversion of fructose to HMF vary between 26 and 92% [90].

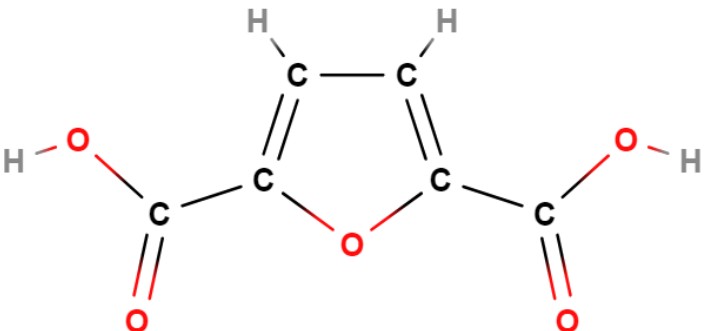

**Figure 11.** Structure of FDCA ($C_6H_4O_5$).

No commercial process has been established for FDCA production from lignocellulosic biomass, and the production of sugar-derived FDCA has only been tested at the pilot scale [91]. Some studies have presented direct conversion (so-called one-pot, two-step method) for 5-HMF synthetized from sugars and oxidation to FDCA [92–95]. Another challenge has been the need for a cost-efficient catalyst for the oxidation of furans, as utilization of noble-metal-assisted catalysts without improved recycling can be costly [89,91]. Zhou et al. [89] developed a partly lignin-derived catalyst, which showed high selectivity, as well as promising substrate conversion and FDCA yields. Some research has been conducted with consideration of enzymatic oxidation, which is suggested as a renewable alternative for catalytic conversion, although the area should be further explored [91,96]. Table 5 provides a summary of the most common oxidation processes used to derive 5-HMF and 2-furoic acid.

**Table 5.** Review of oxidation processes from sugar-derived 5-hydroxymethylfurfural (5-HMF) and 2-furoic acid to 2,5-furandicarboxylic acid (FDCA).

| Substrate | Catalyst | Reagents | Temp. (°C) | Time (h) | Pressure (Bar) | Substr. Conv. (%) | FDCA Yield (%) | Ref. |
|---|---|---|---|---|---|---|---|---|
| 5-HMF | Lignin-derived Co SAs/N@C | $Na_2CO_3$, $O_2$ | 85 | 3 | 1 | 99.4 | 74.4 | [89] |
| | | | 85 | 8 | 1 | 100 | 99.5 | |
| | $MnO_2$ | $NaHCO_3$ | 100 | 24 | 10 | >99 | 91 | [97] |
| | $Au-TiO_2$ | N/A | 65 | 8 | 10 | N/A | >99 | [98] |
| | 5% Pt/C | $O_2$ | 100 | 20 | 40 | N/A | 94 | [92] |
| | Magnetic $ZnFe_{1.65}Ru_{0.35}O$ | Dimethyl sulfoxide | 130 | 16 | N/A | N/A | 91.2 | [94] |
| | Ru/HAP | N/A | 160 | 4 | 20 | N/A | 34.2 | [94] |
| | Pd/CC | $K_2CO_3$, $O_2$ | 140 | 30 | N/A | N/A | 85 | [95] |
| | Ru (4%)/$MnCo_2O_4$ | N/A | 120 | 10 | 24 | 100 | 99.1 | [99] |
| | Fungal enzymes: aryl alcohol oxidase, peroxygenase, galactose oxidase | $H_2O_2$, phosphate buffer | N/A | >24 | N/A | N/A | 80 | [96] |
| 2-furoic acid | N/A | $Cs_2CO_2$, $CO_2$ | 200 | 5 | 8 | N/A | 77 | [88] |
| | Lignin-derived Co SAs/N@C | $Cs_2CO_2$, $CO_2$ | 260 | 36 | Flowing | 85.8 | 71.1 | [89] |

### 3.11. Xylitol

Xylitol is featured as a top value-added product from biorefineries in both reports published by the USDOE and is one of the most studied molecules due to its applications in the pharmaceutical, cosmetic, and food industries [100]. The Asia Pacific region (China

in particular) represents a disproportionally large share of xylitol as compared to the rest of the world, with chewing gum being the major market, representing 80–90% of the total demand in Asia [100,101]. Xylitol demand grew from 6000 tons in 1978 to 190 thousand metric tons in 2016, which was valued at USD 725.9 million [102].

Xylitol (structure in Figure 12) can be derived through the hydrogenation of xylose-by-xylose reductase; however, microbial production of xylitol using yeast fungi and bacteria has proven to be the more promising route for production. *Candia* spp. is the most studied fungus with respect to xylitol production [103]. Xylitol serves as a precursor for the production xylaric acid, ethylene glycol, and propylene glycol. Table 6 summarizes reported xylitol production from lignocellulosic biomass. Whereas the pretreatment conditions are quite diverse, the hydrolysate concentration is a good indicator of the relative effectiveness of pretreatment.

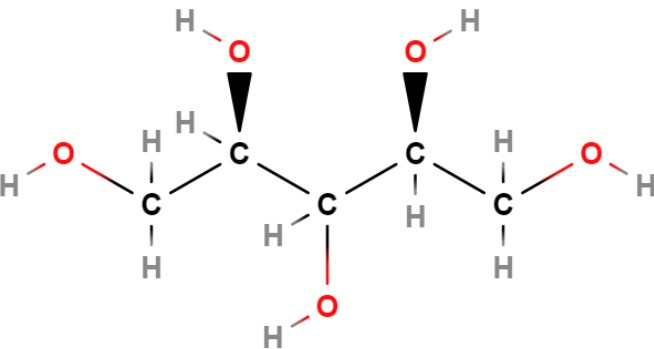

**Figure 12.** Structure of xylitol ($C_5H_{12}O_5$).

### 3.12. Chitosan

Chitosan (deacetylated chitin) is biopolymer used in food, pharmaceutical, and cosmetics industries due to its non-toxicity, biocompatibility, biodegradability, and antimicrobial properties [104,105]. It mainly consists of amino sugar D-glucosamine bounded to small amounts of N-acetyl-D-glucosamine [105]. Satari et al. [106] approximated the amount of chitosan in a biomass sample by measuring the amount of D-glucosamine. A representative structure of chitosan can be seen in Figure 13.

Traditionally, chitosan is produced from waste exoskeletons of shellfish, but as a high concentration (45–60%) NaOH is needed to extract the chitosan from chitin, fungus-derived chitosan is a suitable alternative, as the more diluted alkaline required for the extraction decreases the environmental pollution caused by the process [105]. Sigma-Aldrich (St. Louis, MO, USA) and ChitoLytic (Toronto, ON, Canada) are companies producing non-animal-derived chitosan at the commercial level. Chitosan can be found in the cell wall of *Zygomycetes* fungi, and strains from *Rhizopus* and *Mucor* genera have been tested for fermentation of lignocellulose prehydrolysates [104–107]. Xylose, which is digested by microorganisms, is released from lignocellulose by treating the biomass using hydrothermal or acid-assisted pretreatment at moderate temperatures, which can be followed by enzymatic hydrolysis [108]. Despite being inhibitors, some sugar degradation compounds, such as formic acid and acetic acid, in moderate concentrations have been shown to stimulate the fungal growth and increase the accumulation of protective chitosan in the fungal cell wall [104]. However, severe pretreatment conditions must be avoided to prevent the excessive production of inhibitory compounds for fungal growth, such as furfural. Chitosan can be extracted from fungal biomass by first separating the alkaline, insoluble material, which is subsequently extracted using diluted acetic acid or acetate [104–107]. Tai et al. [104] demonstrated fermentation in hemicellulose-based hydrolysate to enhance the fungal growth and chitosan production compared to fermentation in synthetic glucose and xylose-containing medium. Table 7 provides a summary of fungal chitosan production from lignocellulosic biomass.

**Figure 13.** Representative structure of chitosan.

### 3.13. 2,3 Butanediol (2,3-BD)

Long-chain alcohol 2,3-butanediol (2,3-BD), structure seen in Figure 14, is an important platform bulk chemical that is used as a fuel additive and in various other industries, including chemicals, plastic manufacturing, pharmaceuticals, cosmetics, and even food [109–111]. Currently, it is mainly sourced from the petrochemical industry, but the increasing interest in biorefining and sustainable bio-based chemicals has made microbial production of 2,3-BD a desirable alternative [112–114].

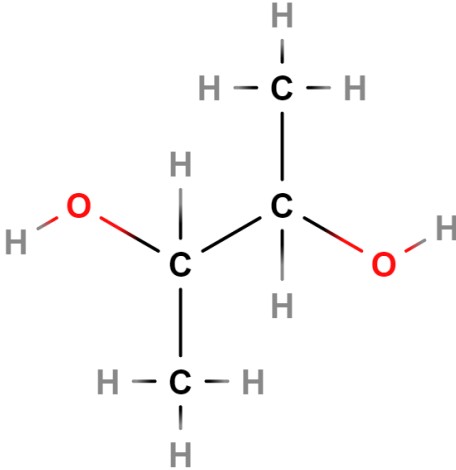

**Figure 14.** Structure of 2,3-BD ($C_4H_{10}O_2$).

2,3-BD can be produced by fermentation of lignocellulose-derived sugars using bacterial strains that can simultaneously utilize glucose and xylose. Many industrial strains can biologically produce 2,3-BD and have been shown to be suitable for fermentation. However, to enhance the production and ensure sufficient yields, metabolic engineering of the strains has been studied [109,110,112,114,115]. The pathogenic nature of most robust strains in *Klebsiella* and *Enterobacter* genera remains an obstacle to commercialization of microbial 2,3-BD production. On the other hand, generally recognized as safe (GRAS) organisms, such strains in the *Bacillus* genus, have shown lower fermentation efficiency [109,115]. Lignocellulose is pretreated and hydrolyzed before fermentation, usually by sodium hydroxide or sulfuric acid treatment followed by enzymatic hydrolysis. Joo et al. [116] studied the effect of inhibitory sugar degradation compounds and reported that formic acid, furans, and phenolic compounds have negative effects on cell growth and 2,3-BD production, which enhances the importance of appropriate pretreatment conditions. Table 8 provides a summary of fermentation techniques used to derive 2,3-BD from lignocellulosic biomass.

### 3.14. Xylo-Oligosaccharides (XOs)

Xylo-oligosaccharides (XOs) are non-digestible carbohydrates with prebiotic and other beneficial health properties that have gained commercial interest due to their potential for use as nutraceuticals; they can be produced by hydrolyzing xylan, the main component of hemicellulose. [117,118]. This provides a route to produce high-value compounds from abundant and inexpensive lignocellulosic feedstock, such as agricultural residues [119,120]. A representative structure of XOs can be seen in Figure 15.

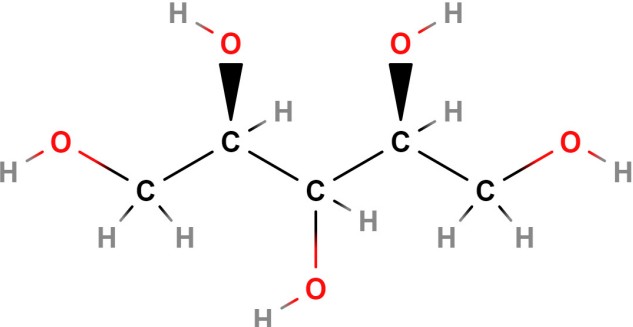

**Figure 15.** A representative structure of XOs $(C_5H_{12}O_5)_n$.

Several methods have been tested for XO production from lignocellulose, the most common of which is hydrothermal or alkaline pretreatment followed by enzymatic or acid hydrolysis. Hydrothermal pretreatment [120], acid hydrolysis [121], and enzymatic hydrolysis [122] without pretreatment have also been tested. However, the high temperatures required for hydrothermal pretreatment and autohydrolysis increase the energy consumption of the process. Therefore, it is suggested that alkaline pretreatment at moderate temperatures, together with enzymatic hydrolysis, could be the most sustainable processing route [118]. Downstream processing is required to purify XO-containing liquors (75–90% purity required for food applications) from unwanted toxic and unhealthy compounds, such as furfural and 5-HMF, and this process directly impacts the production costs [118]. Production of XOs from agricultural residues has been extensively studied in recent years, and in the reviewed studies, the reported yields, depending on the xylan extraction and hydrolysis method, vary between 10.2% from wheat straw [119] up to >99% from sugarcane bagasse [123]. One of the most promising feedstocks is corn cob, which has shown high yields in various studies [124–127]. Table 9 summarizes XO production from lignocellulose.

**Table 6.** Review of extraction methods for xylitol from lignocellulosic biomass.

| Biomass | Feedstock Content | Acid Conc. | Operating Temp. (°C) | Time (min) | Hydrolysate | Strain | Xylitol Reported | Reference |
|---|---|---|---|---|---|---|---|---|
| Corncob | 20.91 g/L of Xylose | 1% $H_2SO_4$ ($v/v$) | 121 | 30 | 40.16 g/L and 52.71 g/L of xylose (hydrolysate concentrated through rotavapor and microwave, respectively) | *Candida tropicalis* | 1.2-fold increase in yield and 1.8-fold increase in productivity | [128] |
| | 31.2 g/L of xylose and 3.3 g/L of glucose | 1% $H_2SO_4$ ($v/v$) | 121 | 40 | 160 g/L of xylose | *Candida tropicalis* As 2.1776 | Yield of 0.83 g/g and productivity of 1.01 g/L/h. Maximum xylitol production of 96.5 g/L | [129] |
| | 41.2% cellulose, 33.4% hemicellulose, and 18.7% lignin | 1% $H_2SO_4$ ($v/v$) | 125 | 60 | 21.67 g/L of xylose | *Candida tropicalis* CCTCC M2012462 | Maximal xylitol concentration of 38.8 g/L. Yield of 0.7 g/g of xylose and a productivity of 0.46 g/L/h | [130] |
| | 42.7% cellulose, 34.3% hemicellulose, and 17.5% lignin | 1% $H_2SO_4$ ($v/v$) | 120 | 60 | 28.7 g/L of xylose | *Candida tropicalis* W103 | Maximal xylitol concentration of 68.4 g/L. Yield of 0.7 g/g xylose and a productivity of 0.95 g/L/h | [131] |
| | 32% cellulose, 35% hemicellulose, 20% lignin, and 4% ash and others | 1% $H_2SO_4$ ($v/v$) | 121 | 60 | 24.9 g/L of Xylose | *Candida magnoliae* | Production rate of 0.51 g/L/h and 18.7 g xylitol/L | [132] |
| Sugarcane bagasse | - | - | - | - | 65% xylose, 15% arabinose, and 8% glucose | *Debaryomyces hansenii* | Maximum yield of 0.76 and 0.82 g/g using free and immobilized cells, respectively, with corresponding volumetric productivities of 0.44 and 0.46 g/L/h at 100 g/L initial xylose concentration | [133] |
| | 17.5% DM loading | 1% $H_2SO_4$ | 150 | 30 | 15.73 g/L of D-xylose | *Candida guilliermondii* FTI 20037 | Maximal xylitol production of 50.5 g/L. Yield of 0.81 g/g of xylose and productivity of 0.6 g/L/h | [134] |

**Table 6.** *Cont.*

| Biomass | Feedstock Content | Acid Conc. | Operating Temp. (°C) | Time (min) | Hydrolysate | Strain | Xylitol Reported | Reference |
|---|---|---|---|---|---|---|---|---|
| Sugarcane bagasse | 9.3 xylose, 15.2 glucose, and 8.5 lignin (% of DM) | 100 mg of sulfuric acid per g of bagasse (dry wt) | 121 | 10 | Hydrolysate 64.7 g/L xylose, 3.08 g/L glucose, 4.23 g/L arabinose, and 1.84 g/ L acetic acid | *Candida guilliermondii* | Maximum xylitol concentration of 28.7 g/L, xylitol yield on consumed xylose of 0.49 g/g, and a xylitol volumetric productivity of 0.24 g/L/h | [135] |
| | 10% DM loading | 1% $H_2SO_4$ (*v/v*) | 121 | 60 | Sugar composition in hydrolysate xylose 56%, glucose 15%, and arabinose 24% | *Candida tropicalis* | Xylitol yield was 0.65 g/g of xylose | [136] |
| Corn Fiber | 20% DM loading | 1% $H_2SO_4$ (*v/v*) | 121 | 60 | Sugar composition in hydrolysate xylose 30%, glucose 38%, arabinose 22%, and galactose 4%. | *Candida tropicalis* | Xylitol yield was 0.65 g/g of xylose | |

DM: dry matter.

**Table 7.** Review of production methods for fungal chitosan from lignocellulose.

| Biomass | Treatment Conditions | Xylose (g/L) | Fungal Strain | Chitosan Extraction | Biomass Production (g/L/day) | Chitosan Content (g/g Biomass) | Comment | Ref. |
|---|---|---|---|---|---|---|---|---|
| Corn Stover | 2% $H_2SO_4$ 100 °C 2 h | 22.4 | *Rhizopus oryzae* ME-F12 | 1 M NaOH at 121 °C for 15 min + 2% acetic acid at 95 °C for 24 h | 5.2 | 0.09 | Total production | [104] |
| | Autoclave at 121 °C for 20 min | N/A | *Aspergillus niger* | 1 M NaOH at 121 °C for 20 min + 2% acetic acid at 95 °C for 6–8 h | 15.8 | 6.8 | (g/kg) Solid-state fermentation | [107] |
| | Autoclave at 21 °C 20 for min | N/A | *Rhizopus oryzae* | 1 M NaOH at 121 °C for 20 min + 2% acetic acid at 95 °C for 6-8 h | 14.6 | 8.6 | | |
| | Acid-assisted steam explosion, 0.8 MPa | 30 | *Rhizopus oryzae* AS 3.819 | 1 M NaOH at 121 °C for 15 min + 2% acetate at 95 °C for 24 h | 3.7 | 0.09 | N/A | [105] |
| Elm Wood | 85% $H_3PO_4$ at 60 ° for C 45 min + enzymatic | ND | *Mucor indicus* CCUG 22424 | 0.5 M NaOH at 121 °C for 20 min (alkali-insoluble material) | 3.3 | 0.06 | Determined as the amount of glucosamine | [106] |
| Pine Wood | 85% $H_3PO_4$ at 60 °C for 45 min + enzymatic | 6.9 | *Mucor indicus* CCUG 22424 | 0.5 M NaOH at 121 °C for 20 min (alkali-insoluble material) | 2.8 | 0.06 | | |
| Rice Straw | 85% $H_3PO_4$ at 60 °C for 45 min + enzymatic | ND | *Mucor indicus* CCUG 22424 | 0.5 M NaOH at 121 °C for 20 min (alkali-insoluble material) | 3.1 | 0.06 | | |
| Wheat Straw | NMMO * 120 °C 3 h + enzymatic | 19.8 | *Mucor indicus* CCUG 22424 | Autolysis + NaOH treatment + extraction | N/A | 0.13 | N-methylmorpholine -N-oxide | [137] |

* N-methylmorpholine-N-oxide.

**Table 8.** Review of fermentation methods for 2,3-butanediol (2,3-BD) from lignocellulose.

| Biomass | Pretreatment | Hydrolysis | Strain | Productivity (g/L/h) | 2,3-BD Conc. (g/L) | Yield (%) | Ref. |
|---|---|---|---|---|---|---|---|
| Corn cob | 2% NaOH at 80 °C for 2 h | Enzymatic | *Enterobacter cloacae* CICC 10011 | 0.9 | N/A | 42 | [138] |
| Corn stover | 0.1 M NaOH at 80 °C | Enzymatic | *Zymomonas mobilis* | N/A | 10 | N/A | [114] |
| | N/A | Enzymatic | *Paenibacillus polymyxa* | 1.1 | 18.8 | 31 | [139] |
| | N/A | N/A | *Bacillus licheniformis* | 2.3 | 119.4 | 95 | [112] |
| Jerusalem Artichoke Stalk | 1% H$_2$SO$_4$ at 130 °C for 90 min | Enzymatic | *Klebsiella pneumoniae* | N/A | 80.5 | 16.8 | [111] |
| Oil Plan Frond | 3% NaOH at 121 °C for 20 min | Enzymatic | *Enterobacter cloacae* SG1 | 0.3 | 30.7 | N/A | [113] |
| Pine Tree | N/A | N/A | *Klebsiella oxytoca* CHA006 | 0.7 | 5.8 | 30 | [110] |
| Rice Straw | 0.375 M NaOH at 120 °C for 20 min | Enzymatic | *Klebsiella* sp. Zmd30 | 2.4 | N/A | 62 | [140] |
| Rice Waste | Na$_2$CO$_3$ + NaHCO$_3$ + Na$_2$SO$_4$ at 100 °C for 3 h | Enzymatic | *Klebsiella pneumoniae* KMK-05 | 0.48 | 11.5 | 38.4 | [141] |
| Sorghum Stalk | 1.25% NaOH at 121 °C for 30 min | Enzymatic | *Bacillus licheniformis* DSM 8785 | 1 | N/A | 45 | [109] |
| | 0.375 M NaOH at 120 °C for 20 min | Enzymatic | *Klebsiella* sp. Zmd30 | 0.7 | N/A | 15 | [140] |
| Sugarcane Bagasse | 10% NaOH at 90 °C for 90 min | Enzymatic | *Klebsiella pneumoniae* CGMCC 1.9131 | N/A | 9 | N/A | [142] |
| | 1% H$_2$SO$_4$ at 121 °C for 30 min | N/A | *Enterobacter aerogenes* EMY-22 | 0.8 | 66.4 | 42 | [110] |
| | 5% Na$_2$CO$_3$ + 5% Na$_2$SO$_3$ at 100 °C for 4 h | N/A | *Enterobacter aerogenes* EMY-22 | N/A | N/A | 39.5 | [143] |
| Sunflower Stalk | N/A | N/A | *Klebsiella oxytoca* CHA006 | 0.8 | 4.3 | 34 | [110] |
| Wood | 24 N H$_2$SO$_4$ at 30 °C for 60 min + diluted acid at 105 °C for 60 min | N/A | *Enterobacter aerogenes* | N/A | 9.9 | N/A | [116] |
| | N/A | N/A | *Bacillus licheniformis* DSM 8785 | 1.6 | N/A | 40 | [109] |

**Table 9.** Review of xylo-oligosaccharide (XO) production from lignocellulose.

| Biomass | Hemicellulose (%) | Xylan (%) | Extraction/Pretreatment | Hydrolysis | XO Conc. (g/L) | Yield | Unit | Ref. |
|---------|-------------------|-----------|--------------------------|------------|----------------|-------|------|------|
| Barley Husk | N/A | 26.8 | N/A | Autohydrolysis at 220 °C for 0.75 h | N/A | 27.1 | % | [120] |
| Corn Cob | N/A | 30.6 | N/A | Non-isotherm autohydrolysis at 202 °C | N/A | 78.7 | gXOs/100 g xylan | [126] |
| | 38.8 | N/A | 12% NaOH + steam at 121 °C for 45 min | 0.25 M $H_2SO_4$ at 90 °C for 60 h | 0.9 | N/A | N/A | [144] |
| | 38.9 | N/A | 4–16% NaOH + steam at 121 °C for 45 min | Enzymatic at 40.9–41.4 °C for 16.6–17.3 h | <2.0 | N/A | N/A | [145] |
| | N/A | N/A | 1.25 M NaOH at 37 °C for 180 min | Enzymatic at 45 °C for 8 h | 6.7 | 60 | % | [124] |
| | N/A | 31.3 | N/A | Autohydrolysis at 220 °C for 0.75 h | N/A | 24.8 | % | [120] |
| | N/A | 31.9 | 2% NaOH at 20 °C for 6 h | Enzymatic at 50 °C for 24–36 h | 8.2 | 86.7 | % | [125] |
| | N/A | 34.8 | 1.0 g/L $H_2SO_4$ + steam at 135 °C for 30 min | Enzymatic at 50 °C for 24 h | N/A | 67.7 | gXOs/100 g xylan | [127] |
| Eucalyptus Wood | N/A | 16.6 | N/A | Autohydrolysis at 220 °C for 0.75 h | N/A | 15.4 | % | [120] |
| Maize Silage | 35.1 | N/A | 1 M NaOH + steam at 121 °C for 15 min | Enzymatic at 50 °C for 24 h | 3.5 | N/A | N/A | [117] |
| Oil Palm Frond | 30.4 | N/A | Steam at 121 °C for 60 min | Enzymatic at 40 °C for 24 h | N/A | 17.5 | *w/w*% | [146] |
| Reed | N/A | 21.2 | Steam at 170 °C for 30 min | Enzymatic at 50 °C for 48 h | N/A | 68.1 | gXOs/100 g xylan | [147] |
| Rice Husk | N/A | 15.6 | N/A | Autohydrolysis at 220 °C for 0.75 h | N/A | 18 | % | [120] |
| | 25 | N/A | 18% NaOH + steam at 120 °C for 45 min | Enzymatic at 50 °C for 9 h | N/A | 34.7 | gXOs/100 g xylan | [148] |
| | 11.2 | N/A | N/A | Enzymatic at 50 °C for 24 h | N/A | 69 | gXOs/100 g xylan | [122] |
| Ryegrass Silage | 36.6 | N/A | 1 M NaOH + steam at 121 °C for 15 min | Enzymatic at 50 °C for 24 h | 2.4 | N/A | N/A | [117] |

**Table 9.** *Cont.*

| Biomass | Hemicellulose (%) | Xylan (%) | Extraction/Pretreatment | Hydrolysis | XO Conc. (g/L) | Yield | Unit | Ref. |
|---|---|---|---|---|---|---|---|---|
| | N/A | 20.6 | Aqueous ammonia + steam at 121 °C for 30 min | Enzymatic at 50 °C for 30 h | N/A | >99 | % | [123] |
| | N/A | N/A | N/A | 0.1% $H_2SO_4$ at 140 °C for 1 h | N/A | 92.28 | gXOs/100 g xylan | [121] |
| Sugar Cane Bagasse | N/A | N/A | 6% Alkaline peroxide at 20 °C for 180 min | Enzymatic at 50 °C for 96 h | N/A | 31.5 | % | [149] |
| | N/A | N/A | 10% Acetic acid + steam at 150 °C for 45 min | Enzymatic at 30 °C for 1.25 h | N/A | 39.1 | gXOs/100 g xylan | [150] |
| | 23.2 | N/A | 12% NaOH + steam at 121 °C for 15 min | Enzymatic at 40 °C for 8 h | 1.72 | N/A | N/A | [151] |
| Sunflower Stalk | N/A | 18.9 | 24% KOH at 35 °C for 120 min | Enzymatic at 40 °C for 24 h | 3.2 | N/A | N/A | [152] |
| | N/A | 19.1 | 24% KOH at 35 °C for 120 min | 0.25 M $H_2SO_4$ at 100 °C for 30 min | N/A | 12.6 | gXOs/100 g xylan | [119] |
| Wheat Straw | N/A | 20.6 | 24% KOH at 35 °C for 120 min | Enzymatic 4 at 0 °C for 24 h | 2.3 | N/A | N/A | [152] |
| | N/A | 20.9 | 24% KOH at 35 °C for 120 min | 0.25 M $H_2SO_4$ at 100 °C for 30 min | N/A | 10.2 | gXOs/100 g xylan | [119] |
| | N/A | N/A | 2% NaOH at 80 °C for 90 min | Enzymatic at 60 °C for 15 h | N/A | 39.8 | gXOs/100 g xylan | [153] |

## 4. Extraction Methods

Physical pretreatment is one of the first steps to open the molecular structure of lignocellulosic biomass. Several physical, chemical, and physiochemical methods have been studied and optimized to provide access to cellulose and hemicellulose by opening up the binding lignin structure [154,155]. Whereas the primary goal of pretreatment is to provide accessibility to cellulose and hemicellulose sugars, with time and increased interest in biomass-derived production of value-added chemicals, molecules that were once considered pretreatment inhibitors, such as HMF and furfural, are now the primary compounds of interest. This newfound interest prioritizes the secondary goal, which is opening the lignin structure. Whereas accessibility to polysaccharides still holds relevance, if high-value compounds can be extracted via low-cost extraction methods, then this step precedes the step of extraction of polysaccharides into monomeric sugars. As a result of this shift, new extractions methods are continually being investigated with respect to their efficacy in obtaining high-value biochemicals. Conventional pretreatment operating conditions, as described in Tables 1–9 are often energy-intensive, requiring reactor temperatures above 150 °C and reactor pressure between 1 and 20 atm and are frequently performed in the presence of a catalyst. Novel extraction methods include less energy-intensive alternatives that allow for recovery of high-value chemicals. The attractive prices and wide-ranging applications of specialty chemicals derived from biomass justify the development of new extraction methods. In the following sections, extraction methods for targeted high-value compounds from lignocellulosic biomass are reviewed. The isolation and encapsulation techniques used to stabilize high-value compounds are also discussed and reviewed.

For efficient extraction of high-value compounds from lignocellulosic biomass, physiochemical pretreatment is followed by extraction methods. The extraction method is governed by the characteristics of the targeted compounds identified for extraction. Some compounds are thermolabile and therefore prone to thermal degradation. Therefore, prolonged extractions using high temperatures should be chosen with caution [156,157]. For a solvent to dissolve the solute, the diffusion, solubilization, and/or transfer are governed by the thermodynamic properties of the solute and the solvent. Solvents are widely used for extraction of targeted molecules, partly due to the simplicity of the method, its scalability, and low cost [158]. Solvents used to extract high-value compounds discussed in this review include water, ethanol (EtOH), methanol (MeOH), ethyl acetate (EtOAc), dichloromethane (DCM), acetone, dimethyl sulfoxide (DMSO), deep eutectic and ionic solvents (DESs), acids and bases, and supercritical $CO_2$ [159–166].

Phenolics are secondary metabolites with a chemical structure comprising one or more aromatic rings attached to one or more hydroxyl groups. Phenolics can be derived from food plants and lignocellulosic plants. With more than known 8000 phenolics, there is certainly a growing interest in the ability to extract these compounds from a variety of biomass sources [158]. In an in silico study, Galanakis et al. [167] investigated the ability of solvents to solubilize phenolics, e.g., hydroxycinnamic acids (HCA), flavonoids, phenolic aldehydes, or hydroxybenzoic acids, all of which are molecules with at least one hydroxyl group (polar group) connected to an aromatic ring (non-polar group). The activity coefficient and polarity, which can be predicted by computer models, were the main parameters used to determine the solubility of a solute in a solvent. The authors stated that different solvents target different phenolic groups [167]. Of the 15 investigated phenolic groups, the solvents with the best properties for extracting phenolics were EtOH, MeOH, EtOAc, and DCM. However, these solvents do not consider the liberation of phenolics from the lignocellulosic matrix.

### 4.1. Hansen Solubility Parameters

Hansen solubility parameters (HSPs) are associated with the method of theoretically calculating a solute's solubility in a solvent based on the thermodynamical properties of dispersion, polarity, and hydrogen bonding. Compounds with similar HSP values have high miscibility, solubility, diffusivity, and affinity for each other. Such compounds have

similar HSP values because they have comparable atomic dispersion forces, molecular dipole forces, and electron exchanges, denoted by $\delta_D$, $\delta_P$, and $\delta_H$, respectively.

A compound is defined as soluble in a solvent if the solvent lies inside in three-dimensional solubility parameter spheroid in the Hansen space [168,169]. Hansen emphasizes the impracticality of using water to predict solubility behaviors with HPS. Water is a small molecule with strong polar interactions, as well as strong hydrogen-bonding and hydrogen-donor capabilities. Therefore, the HPS values depend on the local environment, and Hansen does not recommend water predictions of solubility with the HPS method [168,169]. Ionic compounds are not well-described by the HSP method. Tables 10 ad 11 consolidate HSP for common solvents and solutes.

**Table 10.** HSP values of solvents as described by Hansen [169].

| Compound | $\delta_D$ | $\delta_P$ | $\delta_H$ | Compound | $\delta_D$ | $\delta_P$ | $\delta_H$ |
|---|---|---|---|---|---|---|---|
| 1,4-Dioxane | 19 | 1.8 | 7.4 | Iso-butanol | 15.1 | 5.7 | 15.9 |
| 1-Butanol | 16 | 5.7 | 15.8 | Methanol | 14.7 | 12.3 | 22.3 |
| 1-Propanol | 16 | 6.8 | 17.4 | Methyl cyclohexane | 16 | 0 | 1 |
| 2-Butanol | 15.8 | 5.7 | 14.5 | Methyl ethyl ketone | 16 | 9 | 5.1 |
| 2-Propanol | 15.8 | 6.1 | 16.4 | Methyl isobutyl ketone | 15.3 | 6.1 | 4.1 |
| Acetone | 15.5 | 10.4 | 7 | Methylene dichloride | 18.2 | 6.3 | 6.1 |
| Acetonitrile | 15.3 | 18 | 6.1 | N,N-dimethyl acetamide | 16.8 | 11.5 | 10.2 |
| Benzene | 18.4 | 0 | 2 | N,N-dimethyl formamide | 17.4 | 13.7 | 11.3 |
| Benzyl Alcohol | 18.4 | 6.3 | 13.7 | n-Butyl acetate | 15.8 | 3.7 | 6.3 |
| Carbon tetrachloride | 17.8 | 0 | 0.6 | n-Heptane | 15.3 | 0 | 0 |
| Chlorobenzene | 19 | 4.3 | 2 | n-Hexane | 14.9 | 0 | 0 |
| Chloroform | 17.8 | 3.1 | 5.7 | n-Nonane | 15.7 | 0 | 0 |
| Cyclohexane | 16.8 | 0 | 0.2 | n-Octane | 15.5 | 0 | 0 |
| Cyclohexanone | 17.8 | 6.3 | 5.1 | n-Pentane | 15.6 | 0 | 0 |
| Decalin (cis) | 18 | 0 | 0 | sec-Butyl acetate | 15 | 3.7 | 7.6 |
| Dichloromethane | 8.9 | 3.1 | 3 | Styrene | 18.6 | 1 | 4.1 |
| Diethyl Ether | 14.5 | 2.9 | 4.6 | Tetralin | 19.6 | 2 | 2.9 |
| Dimethyl Phthalate | 18.6 | 10.8 | 4.9 | Tetramethylene sulfoxide | 18.2 | 11 | 9.1 |
| Dimethyl Sulfoxide | 18.4 | 16.4 | 10.2 | Toluene | 18 | 1.4 | 2 |
| Ethanol | 15.8 | 8.8 | 19.4 | Water | 18.1 | 17.1 | 16.9 |
| Ethyl Acetate | 15.8 | 5.3 | 7.2 | Xylene | 17.6 | 1 | 3.1 |
| Ethyl Benzene | 17.8 | 0.6 | 1.4 | $\gamma$-Butyrolactone | 19 | 16.6 | 7.4 |
| Ethylene Carbonate | 19.4 | 21.7 | 5.1 | | | | |

Units are in MPa$^{1/2}$.

**Table 11.** HSP values of solutes [169–171].

| Solute | $\delta_D$ | $\delta_P$ | $\delta_H$ |
|---|---|---|---|
| Lactic acid | 17.0 | 8.3 | 28.4 |
| Adipic acid | 17.1 | 9.0 | 14.6 |
| Vanillin * | 19.4 | 9.8 | 11.2 |
| Furfural | 18.6 | 14.9 | 5.1 |
| Ferulic acid * | 19.0 | 6.6 | 15.1 |
| 4-Hydroxy cinnamic acid * | 19.1 | 6.7 | 15.9 |
| Chitosan | 23.0 | 17.3 | 25.7 |
| Xylo-oligosaccharides | 25.4 | 7.4 | 15.5 |

Units are in MPa$^{1/2}$. * Phenolic compound.

By applying the solvent HSP values to the solute HSP values using Equation (1), the distance in Hansen space ($R_{AB}$) can be calculated to determine the theoretically best solvents for individual solutes of interest, as shown in Table 12, where the solvents with the lowest distance to the solute in the Hansen space as defined by $R_1$, the second-lowest is defined by $R_2$, etc.

$$R_{AB} = \sqrt{4 \cdot \Delta\delta_D^2 + \Delta\delta_P^2 + \Delta\delta_H^2} \tag{1}$$

**Table 12.** Chosen solutes of interest and the solvents closest to them in Hansen space.

| Solute | $R_1$ | $R_2$ | $R_3$ | $R_4$ |
|---|---|---|---|---|
| Lactic acid | MeOH | EtOH | 1-Propanol | 2-Propanol |
| Succinic acid | EtOAc | n-Butyl acetate | Sec-Butyl acetate | Chloroform |
| Adipic acid | | | | |
| Vanillin * | Tetramethylene sulfoxide | Benzyl alcohol | N,N-dimethyl acetamide | N,N-dimethyl formamide |
| Furfural | γ-Butyrolactone | Dimethyl phthalate | Dimethyl sulfoxide | Tetramethylene sulfoxide |
| Ferulic acid * | Benzyl alcohol | 1-Butanol | 1-Propanol | 2-Butanol |
| 4-Hydroxy cinnamic acid * | Benzyl alcohol | 1-Butanol | 1-Propanol | 2-Propanol |
| Chitosan | Water | MeOH | EtOH | Dimethyl sulfoxide |
| Xylo-oligosaccharides | Benzyl alcohol | 1,4-Dioxane | Tetramethylene sulfoxide | Dichloromethane |

$R_1$ denotes the shortest distance between solute and solvent in Hansen space, $R_2$ denotes the second shortest distance, etc. * Phenolic compound.

As shown in Table 12, the best solvent for chitosan is water, according to calculations based on HSP values. However, this does not mean that chitosan is soluble in water, as chitosan is a polymer, and a second chemical is therefore needed to change the ionic charge of the solute and solvent, e.g., by the addition of a weak acid [172].

If a solvent with good HSP values compared to the compound of interest is unfit for handling due to safety concerns, high cost, or due to processing inability or environmental restrictions, other miscible solvents with desirable characteristics can be chosen and mixed in ratios that will result in similar HSP values. If mixed on the basis of a volume-weighted average, a new $R_{AB}$ can be calculated such that the mixture of solvents might be low-cost, safer, or more environmentally accepted, with a possibly lower $R_{AB}$ distance in Hansen-space. By applying Equations (2)–(4), new HSP values can be calculated for a mix of solvents. This technique can also shift the solubility of a solute in the mix of solvents to induce crystallization of the solute without extensive and expensive downstream processing.

$$\delta_D = \frac{\delta_{D1} \cdot V_1 + \delta_{D2} \cdot V_2 + \delta_{Dn} \cdot V_n}{V_{tot}} \tag{2}$$

$$\delta_P = \frac{\delta_{P1} \cdot V_1 + \delta_{P2} \cdot V_2 + \delta_{Pn} \cdot V_n}{V_{tot}} \tag{3}$$

$$\delta_H = \frac{\delta_{H1} \cdot V_1 + \delta_{H2} \cdot V_2 + \delta_{Hn} \cdot V_n}{V_{tot}} \tag{4}$$

*4.2. Solvent Extraction*

Maceration extraction (ME) is the most common and easy solvent extraction technique. This method is often called simple solvent extraction or conventional extraction [173,174]. Compounds can be extracted by simply submerging the biomass in a solvent and heating below the boiling point. As ME relies on the diffusion transfer of compounds, physical pretreatment is often required for efficient extraction to increase the surface area of the biomass and possibly to open the plant cell walls of lignocellulosic material. Such physical pretreatment can be achieved by milling or crushing. ME can also happen before, during, or after fermentation. This is the case for winemaking, where the solvent slowly changes from sugar-rich water to a mix of water–sugar–ethanol, shifting the solvent properties, e.g., polarity, dielectric constant, and surface tension [175]. As many organic solvents are perfectly miscible with water, the effect of shifting solvent properties can be well-controlled.

Decoction extraction (DE) involves boiling the plant material in a solvent to extract compounds of interest. This method is also commonly referred to as hot maceration or boiling maceration. Silva et al. [176] investigated DE and microwave-assisted extraction of bioactive compounds from lignocellulosic halophyte *Salicornia ramosissima*. The biomass was milled to open the lignocellulosic structure, and particles of 1 mm were obtained. DE was conducted by boiling 300 mg biomass in 10 mL distilled water for 5 min and leaving it to cool for 25 min. The extract was filtered and freeze-dried into a powder. The extracts

were analyzed by a Folin–Ciocalteu total phenolic assay, with DE showing 80% higher extraction of total phenolics compared to microwave-assisted extraction.

Most analysis methods require liquid samples, as analytical equipment cannot handle solid samples. For analysis, a solid–liquid extraction method is needed to solubilize the solute of interest. Soxhlet extraction is a commonly used method for solid–liquid extraction, as the method is well-described, reliable, safe, and easy to operate. Soxhlet extraction works on the principle of continuous evaporation and simultaneous condensation of a solvent. The biomass is gradually submerged in the solvent. Once the extraction chamber containing the biomass is full of condensed solvent at near-boiling temperature, the solvent is siphoned off, and a new condensed solvent can fill up the extraction chamber. The transfer equilibrium is thereby shifted and does not determine the mass transfer of the desired compound into the solvent. This allows for an unmonitored operation of the Soxhlet apparatus, and the operator can stop the operation when desired. The continuous and cyclic nature of this extraction method ensures a concentrated extract. The solvent is usually determined by the polarity of the solute, but other factors, such as flammability, toxicity, and price, can be used as parameters for the selection of a solvent [177].

Soxhlet extraction can also be applied for the extraction of oils, fats, waxes, sterols, and other non-polar compounds from lignocellulosic biomass using a non-polar solvent. More polar compounds, e.g., phenolic compounds, can be extracted using ethanol, methanol, or ethyl acetate. A schematic of this setup is shown in Figure 16. Cascade extraction using a Soxhlet apparatus can achieve relatively pure extract phases and yield a lignocellulosic fraction with a low concentration of residual material, such as lipids or ash [178–180].

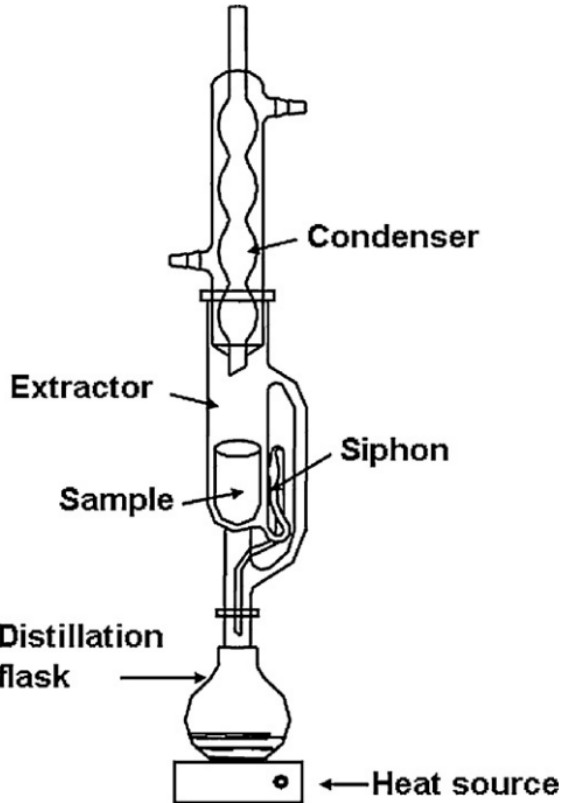

**Figure 16.** Experimental setup for experimental Soxhlet extraction. Adapted from [177].

Medini et al. [178] studied the phytochemical composition, antioxidant capacity, anti-inflammatory effect, and anticancer activities of the halophyte *Limonium densiflorum*. The tested extracts were extracted by Soxhlet extraction using a successive cascade with hexane, dichloromethane, ethanol, and methanol. These solvents were carefully chosen to extract as much non-lignocellulosic material as possible and thereby afford a pure lignocellulosic fraction after extraction [178].

### 4.3. Subcritical Water Extraction

Subcritical water extraction (SWE) utilizes increasing temperatures and pressure of water of 100–374 °C and 0.1–22.1 MPa, respectively, while remaining below the vapor pressure. SWE utilizes the shift in chemical properties, such as viscosity, dielectric constant, surface tension, diffusivity characteristics, polarity, and pH value. Changing these properties can cause the water assimilate acids and organic solvents, allowing the operator to select the compounds of interest and yielding an aqueous extract free of harmful acids and solvents [181]. Zhang et al. [181] reviewed the extraction of phenolic compounds by SWE and found the optimal extraction conditions for total phenolic content (TPC) and total flavonoid content (TFC) of lignocellulosic biomass to be approximately 160–220 °C for 16–45 min at 2–6 MPa. Table 13 summarizes the parameters and raw materials reviewed by Zhang et al. [181]. The optimized extraction parameters are changed by increasing the pressure, and the extraction temperature can be lowered significantly to 100–140 °C for 5–130 min at 10–15 MPa.

**Table 13.** Optimal extraction parameters by subcritical water extraction measured by Folin–Ciocalteu total phenolics assay. Literature reported in a review by Zhang et al. [181].

| Detection Method | Raw Material | Temperature (°C) | Pressure (MPa) | Time (min) | Ref. |
|---|---|---|---|---|---|
| | *Oleaceae europaea* pulp residue | 160 | - | 30 | [182] |
| | *Vitis vinifera* pomace | 140 | 15 | 130 | [183] |
| | *Vitis vinifera* pomace | 120 | 10 | 120 | [184] |
| Folin–Ciocalteu | *Plantago major* | 100 | 10 | 2 | [185] |
| | *Plantago lanceolata* | 100 | 10 | 2 | [185] |
| | Japonica-type rice (*Oryza sativa*) | 100–360 | 18 | 10–30 | [186] |

### 4.4. Extraction at Varying pH Values

Changing pH during an extraction process can purify the product from other compounds that will dissolve in acidic or alkali solutions and hence extract unwanted compounds. Examples include polymers and resin chitosan, 2,5-furandicarboxylic acid polymers, and furfural resins.

Treating biomass using alkalis is a common method for pretreatment or extraction of certain compounds [85,187]. To avoid irreversible degradation of many compounds with low $pK_a$ values, such as organic acids, the stabilities of compounds of interest should be reviewed. Friedman et al. [187] demonstrated the irreversible degradation of caffeic, chlorogenic, and gallic acids at high pH, and degradation was shown to occur even at acidic pH levels, which emphasizes the necessity of low pH values during storage and extraction of phenolic compounds. Chlorogenic acids were shown to degrade into their subcomponent monophenolic compounds, caffeic acid and quinic acid. Flavonoids were shown to be more stable than monophenolic compounds or heterodimers, such as chlorogenic acid. Friedman et al. explained that this was due to the exposed carboxyl groups on the less complex phenolic compounds [187]. However, Peanparkdee et al. contested that protocatechuic acid, similar in molecular structure to gallic acid and thus considered to be more stable, exhibited degradation at lower pH levels [188]. Table 14 summarizes the stability and degradation of phenolic compounds at varying pH.

**Table 14.** Review of stability and degradation of phenolic compounds at varying pH levels.

| Compound | Stable at pH | High Degradation at pH | Refs. |
|---|---|---|---|
| Gallic acid | <7 | >10 | [187] |
| Protocatechuic acid | <5 | >7 | [188] |
| Vanillic acid | <5 | >7 | [188] |
| Caffiec acid | <7 | >10 | [187] |
| Ferulic acid | <9 | >11 | [187,188] |
| Quercetin | N/A | >6 | [53] |
| (-)-catechin | <7 | >10 | [187] |
| Rutin | <11 | N/A | [187] |
| Neochlorogenic acid | <6 | >9 | [189] |
| Cryptochlorogenic acid | <5 | >9 | [190] |
| Chlorogenic acid | <5 | >7 | [191,192] |

*4.5. Ultrasound Extraction*

Ultrasound extraction (USE) has been studied extensively with the purpose of providing enhanced accessibility to cellulose and hemicellulose. Bussemaker and Zhang [193] reviewed the application of ultrasound on various lignocellulosic biomasses, such as sorghum, corn stover, sugarcane bagasse, rice hull, and rice straw. However, the focus, like that of other reviews, was the extraction of lignin from biomass or the extraction of cellulose or hemicellulose [194–196]. Fang et al.'s [197] book, '*Production of biofuels and chemicals with ultrasound*' covers the extraction of biofuels and enhanced biogas production using ultrasound, as well as the extraction of chemicals from algae. The use of ultrasound to extract value-added chemicals from lignocellulosic biomass remains scarcely researched. This expands the review slightly to accommodate other biomasses that are not necessarily lignocellulosic but have been utilized for the extraction of high-value chemicals using ultrasound.

Corbin et al. [198] investigated the USE of flax seeds for efficient extraction of phenolic compounds. The phenolics extracted by Corbin et al. [198] were bound to a glucoside group, as the phenolics in flaxseed are bound in the seed coat, with high content of glucosidic bonds. The molecules are still considered HCA, despite the glucosidic bonds. Corbin et al. [198] used slightly alkaline operation conditions (0.2 N NaOH) for the compounds bound in the seed coat matrix to release the polyphenolic lignans, polyphenolic flavonoids, and monophenolic hydroxycinnamic acids. The study by Corbin et al. [198] also compared phenolic extractions in flax seeds by optimized microwave extraction (MWE), USE, enzymatic-assisted extraction (EASE), and heated reflux. This shows MWE to be superior for the extraction of ferulic acid glucoside (76 w% higher than USE), USE to be superior for the extraction of *p*-coumaric acid glucoside (20 w% higher than MWE), and MWE and USE to show similar properties in the extraction of caffeic acid glucoside. The extraction of other polyphenolic compounds was also dominated by MWE and USE extractions in flaxseed, with USE showing superior extraction properties [198].

USE has also been investigated with respect to the extraction of phenolics from saline lignocellulosic biomass and halophytes. Padalino et al. [199] extracted phenolics from vacuum-dried fresh *Salicornia europaea* to increase the phenolic content and antioxidant capacity of freshly made pasta. USE was executed at an extraction temperature of 50 °C with an ethanol/water ratio of 40/60 *v*/*v*% and a DM loading of 1:30 *w*/*v*. The researchers achieved an increase in antioxidant capacity of 148% [199].

*4.6. Enzymatic Extraction*

Arabinoxylan–lignin and glucan–lignin linkage with the HCAs ferulic acid, *p*-coumaric acid, and flavonoid tricin, amongst others, were shown to be present in wheat straw by Zikeli et al. [200]. Of the carbohydrate–lignin linkages in wheat straw, the ferulic acid linkage is the most predominant, with high amounts of tricin linkages also shown between glucan and lignin. These intermolecular bonds are too strong to be broken solvents; hence,

enzymatic hydrolysis could be investigated as part of an extraction cascade, as different extraction methods and solvents have been shown to target the extraction of different phenolics [161,167,198].

Zhu et al. [161] demonstrated that *p*-coumaric acid, ferulic acid, and caffeic acid in dehulled barley cannot be (or are poorly) extracted by regular solvent extraction using acetone but very easily extracted after digestion in 2 M NaOH in the presence of nitrogen. It should be noted that Zikeli et al. [200] found both ferulic acid and *p*-coumaric acid in the carbohydrate–lignin linkage, which is supported by Bartolomé and Gómez-Cordovés' [201] characterization of the purified enzymes from microbial cultures, ferulic acid, and *p*-coumaric acid esterases. Extraction of barley using enzymatic digestion with pepsin, pancreatin, Pronase E, and Viscozyme L, compared to solvent extraction of free and bound phenolics, increased the extraction of (+)-catechin by 232–239 wt% and that of *p*-coumaric acid 29–82 w%, with no significant increase in the extraction of ferulic acid or caffeic acid [162].

Torres-Mancera et al. [202] described enzymatic extraction methods, whereby the majority of the phenolics in ground coffee pulp were recorded to be bound in the plant cell wall. Pectinase was used as a coenzyme to break the structure of the cell wall, and *Rhizomucor pusillus* strain 23aIV was used in solid-state fermentation to extract the HCAs. The phenolics were extracted downstream with solvents [202].

In the case of high protein content in the lignified biomass, protein removal should be considered, as high protein content can inhibit enzymatic hydrolysis, as described by Faulds et al. [203]. For low-protein biomasses, a hydrothermal pretreatment with a low severity factor, followed by enzymatic hydrolysis using commercial enzymatic blend DEPOL 740 L containing ferulic acid and *p*-coumaric acid esterases and subsequent Soxhlet extraction, can be considered. Table 15 describes extraction methods for phenols and hydroxycinnamic acids from lignocellulosic biomass.

**Table 15.** Review of extraction methods for phenolics.

| Biomass | Method | Optimal or Experimental Conditions | References |
|---|---|---|---|
| Flax | USE | 0.2 M NaOH in water at 25 °C for 60 min at 30 kHz | [198] |
| *Crithmum maritimum* and *Salicornia europaea* | USE | Water:ethanol, 40:60 *v/v*% at 50 °C for 20 min | [199,204] |
| Wheat straw | Solvent | Water:ethanol, 60:40 *v/v*%, 8 w% NaOH at 70 °C for 18 h | [200] |
| Barley straw | Alkaline + solvent | Pretreatment: 2 M NaOH for 1 h, nitrogen atmosphere. Solvent: EtOAc. | [161,162] |
| | Enzymatic | Pepsin, Pancreatin, Pronase E, Viscozyme L | |
| Used coffee bean pulp | Enzymatic + fungi + solvent | Pectinase, *Rhizomucor pusillus*, and EtOAc | [202] |
| Brewer's spent grain | Enzymatic | DEPOL 740 L, pH 8 at 50 °C | [203] |

## 5. Isolation and Purification Methods

Once the value-added compounds have been extracted from the biomass, the next step is to isolate and purify said compounds. Here, we discuss membrane filtration, liquid–liquid extraction, and purification using preparative high-performance liquid chromatography (Prep-HPLC).

### 5.1. Membrane Filtration

Membrane filtration can be seen as an easily scalable and inexpensive method for the filtration of extracts. Amongst the various membrane filtration technologies, micro-(MF), ultra-(UF), and nano-(NF) filtration are pressure-driven technologies. These technologies have the benefits of a low energy input, high separation efficiency, simple operation, no use of expensive solvents or effluents, and scalability [205]. A disadvantage of membrane separation is the inability to separate specific compounds of similar polarities and molecular

weights (MWs), as the membranes only retain compounds above a certain molecular size or approximate MW.

Galanakis and Castro-Muñoz et al. [206–208] reviewed the separation of functional macro- and micromolecules using ultra- and nanofiltration (NF). NF with a pore size of 120 Da was used to separate phenolic compounds, achieving a separation efficiency of 99%. The smallest possible phenolic acid, benzoic acid, has an MW of 122 Da, which means all phenolics should be retained by the 120 Da membrane filter. Most phenolics with three or fewer aromatic rings have an MW of <650, which means that an initial filtration to remove larger molecules and particles, such as bacteria, hemicellulose, cellulose, lignin, proteins, and starch, is necessary, as such particles would clog the NF membrane. Even if the range of MW is established as 141–650 Da for membrane separation of phenolic compounds, this does not imply that the molecules will not be retained in larger pores. Galanakis et al. [206] investigated the separation of the phenolic compounds of HCA derivatives and flavanol from olive mill wastewater using UF and NF. Four UF pore sizes and one size of NF (100, 25, 10, 2 kDa, and 120 Da, respectively) were found to retain <1, 32, 44, 53, and 99% HCA, respectively, and 10, 37, 56, 62, and 99% flavonols, respectively [206]. This indicates the ability of these molecules to attach to other molecules or in lignocellulosic structures. Furthermore, 79, 98, 98, 99, 99% of pectin was retained with a pore size of 100, 25, 10, 2 kDa, and 120 Da, respectively, indicating that the optimal UF pore size for retention of pectin and purification of a phenolic-rich fraction is between 25 and 100 kDa, with a secondary NF membrane filtration with a pore size of 120 Da.

Whereas membrane filtration is one of the most promising technologies for recovery of macro- and micromolecules derived from lignocellulosic biomass, it is associated with some drawbacks. Some of the major drawbacks of membrane filtration are membrane fouling, leading to a decrease in permeate flux, as well as reduced efficiency of the process, and the high cost associated with cleaning and maintenance of membranes.

*5.2. Liquid–Liquid Extraction*

Many organic solvents used for liquid–liquid purposes have lower relative polarity than water, and these might coextract lipids, waxes, and other non-polar compounds, along with the targeted compounds. Therefore, a prior liquid–liquid extraction using a non-polar non-water-soluble solvent, such as hexane, dichloromethane, or chloroform, can positively affect the isolation and purification of the targeted compounds, yielding a more concentrated product with fewer contaminates.

Stiger-Pouvreau et al. [209] originally developed a cascade liquid–liquid extraction for the isolation of phlorotannin, a specific phenolic compound group in macroalgae *Sargassacaea* spp. crude extract (Figure 17). A hydroethanolic maceration obtained the extract (*v*/*v*, 1:1). The method implies the use of various consequent washings with organic solvents by liquid–liquid extraction in a separatory funnel, evaporation of solvents, and resuspension of the extracted material in an aqueous phase. Liquid–liquid extraction using dichloromethane extracted the lipids from the crude aqueous extract. Sugars and proteins were separated by low-temperature acetone and ethanol washings. Liquid–liquid extraction using ethyl acetate purified the phenolic compounds and isolated them in the organic solvent [209]. Kim et al. [210] resuspended a powdered methanolic extract of *Salicornia herbacea* in water and successively partitioned it with n-hexane, chloroform, EtOAc, and n-butanol. These compounds are immiscible with water or have a low solubility in water, which can be further decreased by introducing salt [210,211]. The solvent layers were easily separated from the aqueous layers and concentrated in a vacuum at 38 °C. Both the ethyl acetate and the n-butanol phases showed high free-radical scavenging activity, and the compound's two layers were further isolated by column chromatography.

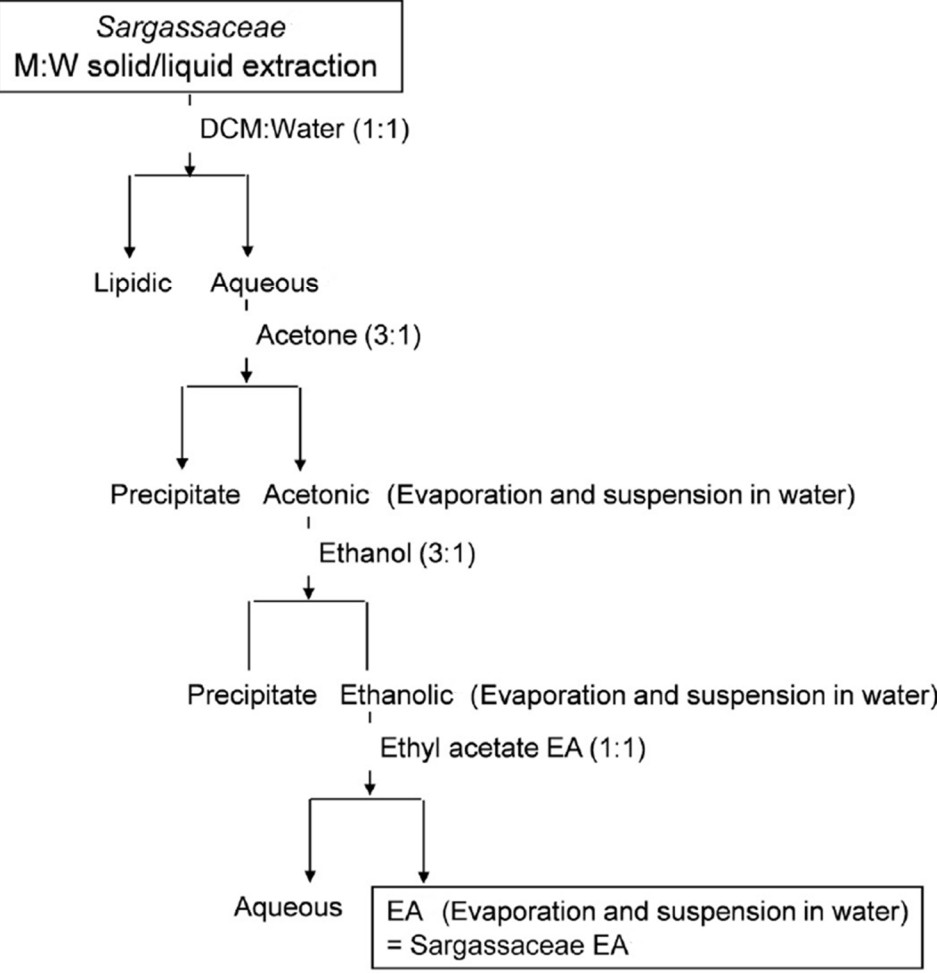

**Figure 17.** Procedure of liquid–liquid extraction of phenolic compounds from *Sargassaceae* spp. [209]. M:W: methanol:water; DCM: dichloromethane.

Similar to membrane filtration, liquid–liquid extraction also has its drawbacks. Bokhary et al. [212] highlighted these challenges, some of which include poor biocompatibility of solvents with microbial species, energy-intensive processes for solvent recovery, solvent toxicity, high cost of solvents, extensive safety protocols, and environmental risks associated with using large quantities of solvents at high temperatures.

*5.3. Purification Using Preparative-HPLC*

Prep-HPLC, like analytical HPLC, is a method of isolating chemical compounds by attracting force retention of compounds in the column stationary phase. While the mixture is passing through the column, the compounds separate in retention time, as detected by ultraviolet (UV) reflection at specific wavelengths, but in prep-HPLC, unlike analytical HPLC, the compounds are quantitatively separated and collected after detection. After collecting the separated compounds, i.e., extract purification, the separated compounds can be further processed and stabilized using encapsulation methods if necessary. Compared to analytical HPLC, Prep-HPLC usually has a larger column diameters and stationary-phase particle diameters, and the operator aims for maximum allowed sample weight, which requires more attention with respect to injection flow rate. To compare the retention times of the separated compounds, the retention factor of compound n ($k'_n$) can be calculated. This is a dimensionless number representing how long a compound is retained in the stationary phase [213–215].

$$k'_n = \frac{t_n - t_0}{t_0} \tag{5}$$



where $t_n$ is the retention time for compound n, and $t_0$ is the column dead time, i.e., the shortest retention time.

Making the peak width as narrow as possible is important to determine the fraction concentration. Unlike a narrow peak with the same area, a broad peak decreases the concentration of the compound of interest, as a more mobile phase dilutes the concentration of the separated compound. The peak width is determined at half the peak height and denoted as $w_{\frac{1}{2}}$. The peak number of compound n ($N_n$) is derived from fractional distillation theory and is a height equivalent to a theoretical plate. The higher the peak number, the narrower the peak in the chromatogram and the higher the concentration of the separated compound [213–215]. The peak number is calculated as:

$$N_n = 5.54 \cdot \left( \frac{t_n}{w_{\frac{1}{2}}} \right) \tag{6}$$

Prep-HPLC can be scaled-up to allow for the separation of larger volumes with a column diameter of up to 30 cm, whereas industrial prep-HPLC can separate up to 3 kg h$^{-1}$ of the analyte, although overloading effects are common. Overloading volume or mass in the column can result in non-optimal peak characteristics, leading to a diluted collected sample, so the maximum injection volume must determined [213,215].

Some of the challenges of using prep-HPLC for large-scale extraction of high-value compounds include the high cost of solvents, challenges in scaling up stationary phase chemistry, and the relationship between the quantity of material recovered and the size of the column [216,217]. However, many such challenges have been successfully addressed in the pharmaceutical sector, which suggests that prep-HPLC is viable for extraction of high-value products [218].

### 6. Conclusions

Research and development activity with respect to the extraction of high-value biomass-derived chemicals have increased considerably since the first report published by USDOE in 2004, highlighting the future of biochemicals. Although the list of top contenders has been altered and appended in the last two decades, several of the top 12 featured compounds have already reached commercial-scale production, such as succinic acid, xylitol, 2,5-FDCA, itaconic acid, levulinic acid, and furfural. Advances in genetically modified microbial strains have also boosted derivation of these compounds using biological routes as an alternative to petrochemical pathways. The considerable advances with respect to processing of lignocellulosic biomasses has enabled researchers to explore cost-effective extraction, isolation, and purification methods that directly target high-value molecules. Whereas pretreatment remains one of the primary steps in opening up the lignocellulosic structure, the severity of pretreatment can be reduced in lieu of the possibility of protecting the unstable phenolic groups, which have a high market value. Although there are still many challenges with respect to the extraction and isolation of high-value compounds, shifting the biorefinery approach to maximize the utility of lignocellulosic biomasses will aid in the cost-effective discovery of novel methods to produce biochemicals from lignocellulosic biomasses.

**Author Contributions:** T.C. and M.H.T. conceptualized and planned the structure of the manuscript. L.S.S.H., M.F. and T.C. performed the literature search and wrote the paper. All authors have read and agreed to the published version of the manuscript.

**Funding:** This review was funded by the Aqua Combine Project of the European Union's Horizon 2020 research and innovation program under Grant Agreement No. 862834.

**Institutional Review Board Statement:** Not applicable.

**Informed Consent Statement:** Not applicable.

**Data Availability Statement:** Not applicable.

**Conflicts of Interest:** The authors declare no conflict of interest.

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
