# Peer review of "Extraction, Isolation, and Purification of Value-Added Chemicals from Lignocellulosic Biomass"

_processes, doi:10.3390/pr10091752_

Round 1
Reviewer 1 Report
Thank you for your breadth of articles and your review overview. I am aware that with such an abundance of different digestion methods, the "depth" is limited. I enjoyed reading about the possibilities for future optimisation via the Hansen parameters. Of course, economic considerations will also play a role, but that is not the task of this article.
Author Response
Thank you for your time and feedback on the manuscript. I appreciate your through review. All errors to language, proof reading, and references have been verified. The references have been cited in the correct manner.
Reviewer 2 Report
The authors compiled an extensive review on extraction, isolation, and purification of value-added chemicals from lignocellulosic biomass. The infromation will be useful for researchers working in the field. However, extensive editing is needed throughout the manuscript. Also, references are not cited properly at many places. Some corrections are marked on the pdf.

Author Response
Thank you for your time and feedback on the manuscript. I appreciate your thorough review. Your comments have been addressed in the attached pdf. All references have been verified and cited in the correct manner.

Reviewer 3 Report
This is a very systematic review on value-added chemicals extraction, isolation and purification from biomass. This fantastic review provided a full picture on biomass related process, with proper details on each value-added chemical production, and very thorough discussion on the purification technology. It would be a very helpful resource for the readers/new researchers who are interested in getting an overview for this field, as well as a nice starting point for anyone who would like to do more research/understand deeper on specific “value-added” chemicals.
A few comments on the structure and format of the paper that the author may consider:
- In Section 2, could the author include an illustration figure to summary different type of value-added chemicals, production process and the market need for the value-added chemicals. The illustration figure will be beneficial for reachers to get the “big picture” of the paper.
- Page2, line 75: should be “value” instead of “vale” (typo)
- In Section 3, could the author include the chemical structures (as a figure) for all the chemicals that discussed in this section.
- In Section 3, could the author change the sequence of the chemicals and the corresponding tables. For instance, currently, table 4 covered section 3.3-3.6, 3.9 table 3 covered section 3.7, table 6 covered section 3.10. This is a little hard for readers to read the content and find related tables.
- In Table 4, please provide the full name of “AFEX”
- Page 20, should be “to open” instead of “to opening”
- In Section 4, the first and second paragraph are the overviews of the extraction method, with brief examples. Thus, the third paragraph on phenolics extraction example could be too detailed. The author may consider move this paragraph to SI.
- Page 21, line 1: “computationally” may not be accurate, could the author use “theoretically” instead?
- Page 22 and 23, some cross-reference on equations were not shown properly “presented as “Error! Reference source not found” in the manuscript. Please fix those.
- Section 4.2, page 23: the last paragraph on using soxhlet extraction in lab-scale (analysis scale) may not be fit for the review, since the main focus of this review is on “large scale” production. The author may consider move it to SI.
- Section 4.4 and 4.5: could the author consider to combine them? Since both are essentially pH driven extraction.
- Section 5, could the author include some discussions on the application challenges for these technology? For instance, membrane filtration performance could be impacted by the membrane pore size distribution, also membrane fouling and scale-up may need to be considered in industrial applications; prep-HPLC regeneration/cycle life should be considered for large scale production.
- Section 5.3: though the calculation derivation on retention factor/peak number/asymmetry of the peak is helpful for understanding the fundamentals of chromatography based separation, however, this could be too detailed for a review. The author may consider move it to SI.
Author Response
Thank you for your time and feedback on the manuscript. I appreciate your through review and your comments have been addressed. Please see my comments below.
A few comments on the structure and format of the paper that the author may consider:
- In Section 2, could the author include an illustration figure to summary different type of value-added chemicals, production process and the market need for the value-added chemicals. The illustration figure will be beneficial for reachers to get the “big picture” of the paper.
This is a good suggestion. I have added Figure 1 describing the biological and chemical routes for the production of top value added chemicals derived from lignocellulosic biomass.
2. Page2, line 75: should be “value” instead of “vale” (typo)
Corrected
3. In Section 3, could the author include the chemical structures (as a figure) for all the chemicals that discussed in this section.
I have added the chemical structures for the discussed chemicals.
4. In Section 3, could the author change the sequence of the chemicals and the corresponding tables. For instance, currently, table 4 covered section 3.3-3.6, 3.9 table 3 covered section 3.7, table 6 covered section 3.10. This is a little hard for readers to read the content and find related tables.
This is a good suggestion and I have rearranged the tables and headings so that they are easier to read.
5. In Table 4, please provide the full name of “AFEX”
Corrected
6. Page 20, should be “to open” instead of “to opening”
Corrected
7. In Section 4, the first and second paragraph are the overviews of the extraction method, with brief examples. Thus, the third paragraph on phenolics extraction example could be too detailed. The author may consider move this paragraph to SI.
Phenolics have not been discussed in the earlier section and is an important compound during extractions, especially for the forthcoming context of HSP. Thus, a small description of phenolics is necessary to better understand the scope and application of the HSP method.
8. Page 21, line 1: “computationally” may not be accurate, could the author use “theoretically” instead?
Corrected
9. Page 22 and 23, some cross-reference on equations were not shown properly “presented as “Error! Reference source not found” in the manuscript. Please fix those.
Corrected
10. Section 4.2, page 23: the last paragraph on using soxhlet extraction in lab-scale (analysis scale) may not be fit for the review, since the main focus of this review is on “large scale” production. The author may consider move it to SI.
This a good suggestion, however the last paragraph of soxhlet extraction technique is trying to highlight teh breadth of solvents available rather than the scale of the extraction itself. Although I agree that review focuses on ‘large scale’ rather than ‘lab scale’, thus that particular sentence has been edited.
11. Section 4.4 and 4.5: could the author consider to combine them? Since both are essentially pH driven extraction.
This is a good suggestion and these sections have now been combined.
12. Section 5, could the author include some discussions on the application challenges for these technology? For instance, membrane filtration performance could be impacted by the membrane pore size distribution, also membrane fouling and scale-up may need to be considered in industrial applications; prep-HPLC regeneration/cycle life should be considered for large scale production.
I have added some challenges associated with scaling all the three techniques (MF, LLE, and prep HPLC) discussed in Section 5
13. Section 5.3: though the calculation derivation on retention factor/peak number/asymmetry of the peak is helpful for understanding the fundamentals of chromatography based separation, however, this could be too detailed for a review. The author may consider move it to SI.
I agree that perhaps the description of the HPLC technique is a bit too detailed and maybe not to most relevant for this review. Thus, I have shortened this section.
Round 2
Reviewer 2 Report
The manuscript can be accepted in present form.